# Sustainable thermal regulation improves stability and efficiency in all-perovskite tandem solar cells

Shuchen Tan [1,3], Chongwen Li [2,3] ✉, Cheng Peng[1], Wenjian Yan[1], Hongkai Bu[1], Haokun Jiang[1], Fang Yue[1], Linbao Zhang[1], Hongtao Gao[1] & Zhongmin Zhou [1] ✉

Mixed Sn-Pb perovskites have emerged as promising photovoltaic materials for both single- and multi-junction solar cells. However, achieving their scale-up and practical application requires further enhancement in stability. We identify that their poor thermal conductivity results in insufficient thermal transfer, leading to heat accumulation within the absorber layer that accelerates thermal degradation. A thermal regulation strategy by incorporating carboranes into perovskites is developed; these are electron-delocalized carbon-boron molecules known for their efficient heat transfer capability. We specifically select *ortho*-carborane due to its low thermal hysteresis. We observe its existence through the perovskite layer showing a decreasing trend from the buried interface to the top surface, effectively transferring heat and lowering the surface temperature by around 5 °C under illumination. *o*-CB also facilitates hole extraction at the perovskite/PEDOT:PSS interface and reduces charge recombination. These enable mixed Sn-Pb cells to exhibit improved thermal stability, retaining 80% of their initial efficiencies after aging at 85 °C for 1080 hours. When integrated into monolithic all-perovskite tandems, we achieve efficiencies of over 27%. A tandem cell maintains 87% of its initial PCE after 704 h of continuous operation under illumination.

Due to their exceptional photoelectric properties and ease of fabrication, the certified power conversion efficiencies (PCEs) of perovskite solar cells (PSCs) have reached 26.1%, matching that of commercial silicon solar cells at a lower cost[1–5]. To further boost the PCEs, mixed tin (Sn)-lead (Pb) perovskites with a more ideal bandgap of around 1.25 eV have garnered increasing attention[6–10]. They are also critical components in all-perovskite tandem solar cells, offering the potential to exceed the Shockley-Queisser limit for single-junction solar cells[11–14]. However, there remains room to further increase their efficiency and stability before achieving requirements for commercialization.

The generation of heat in solar cells caused by (1) energy dissipation as phonons during thermal relaxation and non-radiative recombination processes and (2) reverse-biases heating (Joule heating)

is detrimental to their stability and efficiency[15]. This is particularly prominent in perovskite-based solar cells: their poor thermal conductivity (in comparison with conventional photovoltaic materials such as silicon) results in heat accumulation and the formation of hotspots in solar cell absorbers, making them extremely prone to thermal degradation[15].

Thermal regulation has been proven important in electronics and silicon solar cells for improving device performance and durability but has received limited attention in mixed Sn-Pb PSCs and tandems[16–18]. Due to the inferior thermal stability and facile oxidation of $Sn^{2+}$ in mixed Sn-Pb perovskites that induce extra surface defects for heat generation during recombination[19,20], it becomes crucial to enhance the thermal conductivity and the heat transfer ability in

[1]College of Chemistry and Molecular Engineering, Qingdao University of Science and Technology, Qingdao 266042, China. [2]Department of Electrical and Computer Engineering, University of Toronto, 35 St. George Street, Toronto, ON M5S 1A4, Canada. [3]These authors contributed equally: Shuchen Tan, Chongwen Li. ✉e-mail: lichongwen270@gmail.com; zhouzm@qust.edu.cn

mixed Sn-Pb solar cells. To achieve this, a heat transfer material possessing high thermal conductivity and stability is necessary to be introduced.

Carboranes are electron-delocalized carbon−boron molecules with significant overlap between atomic orbitals of neighboring atoms, allowing electrons to move freely, and spread out over a larger region within the materials[21]. The characteristic of electron delocalization enables them to improve thermal conductivity and efficient carrier

transport[22]. Herein, we selected aromatic *ortho*-carborane (*o*-CB, $C_2B_{10}H_{12}$) as representative due to its exceptional heat transfer ability and chemical stability in the series (Supplementary Fig. 1 and Fig. 1a). We introduced *o*-CB into the precursor solutions of mixed Sn-Pb perovskites and found that *o*-CB exits through the absorber layer showing a decreasing trend from the buried interface to the top surface (Fig. 1b and Supplementary Fig. 2), providing an overall improvement in heat dissipation. In addition, *o*-CB isolates the direct contact between the

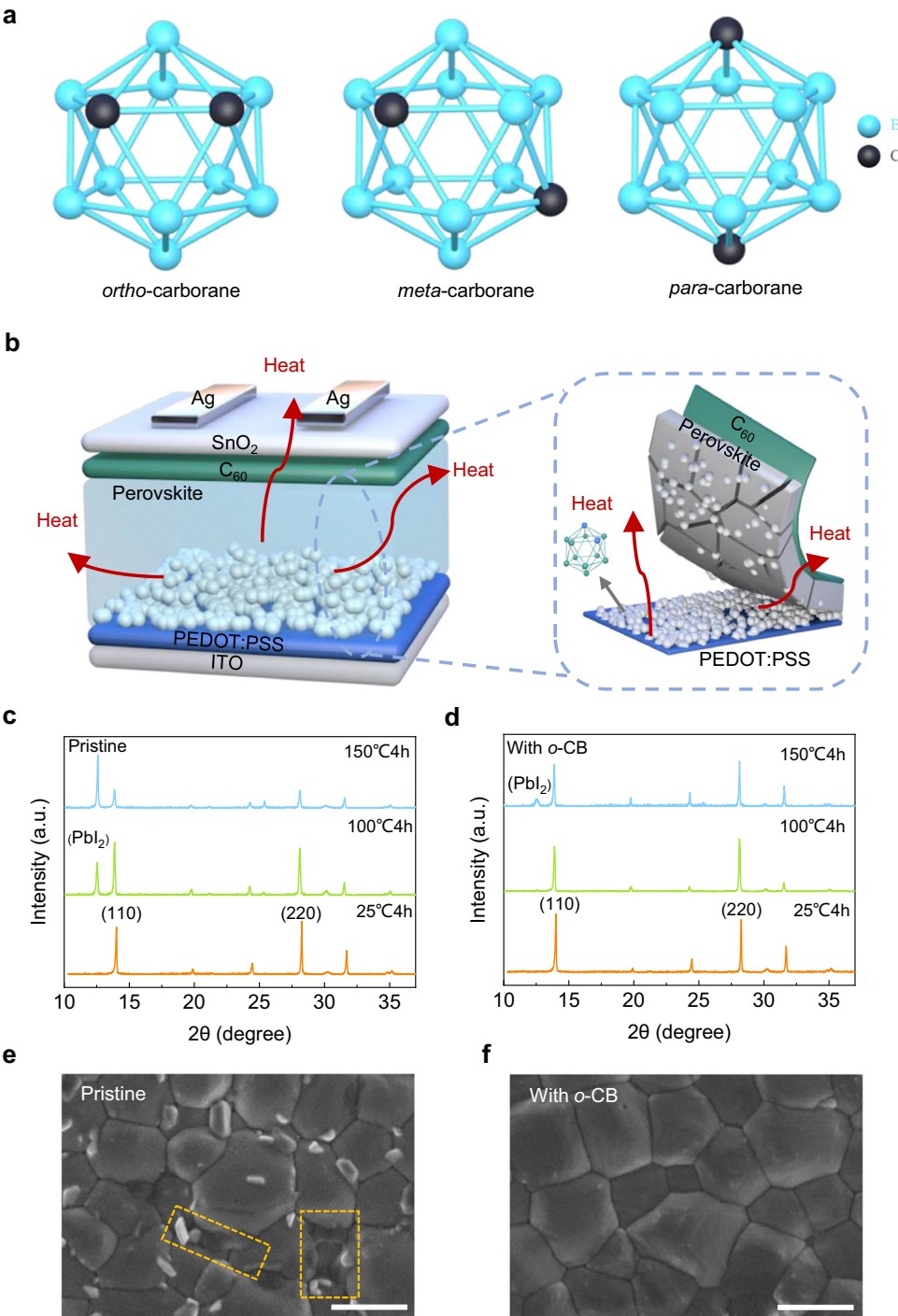

**Fig. 1 | The structure of carboranes and stability of the perovskite films. a** Three Isomers of Carboranes. The smaller of thermal hysteresis (TH), the greater of energy loss, indicating the better heat dissipation. These isomers have the THs of 8 K, 9 K, and 11 K for *ortho*-carborane, *meta*-carborane, and *para*-carborane, respectively. Therefore, *o*-CB has the best heat dissipation ability among these three isomers. **b** Configuration of the p-i-n device treated with *o*-CB and schematic diagram of the heat dissipation. XRD spectra of pristine perovskite films **c** and perovskite films treated with *o*-CB **d** at different temperatures. SEM of the pristine film **e** and treated with *o*-CB **f** after heating at 85 °C for 200 h. The scale bar is 500 nm.

perovskite and the acidic PEDOT: PSS, potentially suppressing the degradation at elevated temperatures.

## Results

To test the improved thermal stability, we first conducted X-ray diffraction (XRD) measurements on perovskite films w/ and w/o *o*-CB after thermal treatment at 25, 100, and 150 °C (Fig. 1c, d). PbI$_2$ peaks were not detected in the original films stored at room temperature. After heating for 4 h at 100 °C, the PbI$_2$ peak intensity of the pristine perovskite film became significantly higher than that of the films treated by *o*-CB. The PbI$_2$ peak intensity of the pristine film further increased after thermal treatment at 150 °C for 4 h, indicating that the pristine perovskite film is less thermally stable. Their degradation was visualized by using scanning electron microscopy (SEM) and the initial morphology of perovskite films w/ and w/o *o*-CB is shown in Supplementary Fig. 3. The *o*-CB treated film showed uniform grains without visible decomposition after heating at 85 °C for 200 h, while second-phase flakes and visible decomposition (yellow rectangle) were observed on the pristine film (Fig. 1e, f), which should be associated with PbI$_2$ residuals that have been detected in XRD.

### Heat transfer

The heat transfer ability was further confirmed by testing the through-plane thermal conductivity (TC) of Sn-Pb perovskite films treated by *o*-CB using the hot-disk technique (see detailed information in the supplementary materials and Supplementary Table 1)[23]. As shown in Fig. 2b, the TC of the pristine perovskite film was 2.476 W m$^{-1}$ k$^{-1}$ at 25 °C. It increased to 2.697 W m$^{-1}$ k$^{-1}$ for the film treated with 1 mg mL$^{-1}$ *o*-CB. The TCs further dropped to 2.441 and 2.592 W m$^{-1}$ k$^{-1}$ at 55 °C, 2.325 and 2.486 W m$^{-1}$ k$^{-1}$ at 85 °C, for pristine and 1 mg mL$^{-1}$ *o*-CB treated films, respectively. This may originate from the reduced electron motion path due to the more frequent collision among lattices at a higher temperature[24]. The improvement in thermal conductivity after *o*-CB treatment is also beneficial for heat dissipation in whole devices as demonstrated in Supplementary Fig. 4.

We used an infrared thermal imaging camera to visually monitor the heat dissipation of mixed Sn-Pb films w/ and w/o *o*-CB[25]. The architecture of the testing films is indium tin oxide (ITO)/PEDOT:PSS/ perovskites w/ and w/o *o*-CB (see detailed information in the supplementary materials). The films were at an original temperature of 85 °C. Once placed on a 25 °C hotplate, the temperature profile of the film with *o*-CB treatment showed faster cooling than the pristine one (Fig. 2c). Besides, the pristine film exhibited consistently higher temperature than the *o*-CB treated sample upon heating (Fig. 2d). The faster cooling and slower heating up in *o*-CB treated samples are attributed to the higher specific heat capacity and lower thermal conductivity of *o*-CB compared to perovskites. We then used T-type thermocouples to conduct real-time temperature monitor to investigate the photothermal response of each layer in PSCs with the architecture shown in Supplementary Fig. 5[26]. The temperature change in each layer in the 2000 s simulated one-sun illumination was analyzed and recorded. The main source of heat generation was found to originate from the perovskite layer. The surface temperature of the pristine film reached 41 °C after 2000 s, while it was 36 °C for the *o*-CB treated sample (Fig. 2e). In addition, we conducted tests on the thermal resistance and thermal transmissivity of the entire device, which provided insights into the device's heat conduction capacity. We can see from Supplementary Table 2 that after the introduction of *o*-CB, the device's thermal resistance at 25, 55, and 85 °C is lower than that of the pristine device at each temperature, while the thermal transmissivity is higher than that of the pristine device at each temperature, thus achieving better heat dissipation.

To gain insight into how the *o*-CB affects the long-term thermal stability of PSCs at high temperatures, the specific heat capacity of *o*-CB under 1 atm at different temperatures was studied. The heat capacity affects the temperature field and, hence, the heat transfer and flow status[27]. The specific heat capacity of *o*-CB increased continuously with temperature (Fig. 2f), resulting in a slowdown of the rate at which the internal temperature rises during heating. In comparison to the pristine film which exhibited a localized area of heat accumulation (hotspot) relating to the temperature of the substrate, the thermal conductivity, and the thickness of the perovskite film[28], the *o*-CB treated film showed improved heat dissipation and a more uniform temperature distribution (Fig. 2g, h and Supplementary Fig. 6).

### Recombination reduction

The existence of *o*-CB doesn't change the perovskite composition (ultraviolet-visible absorption spectra and XRD, Supplementary Figs. 7 and 8), but modifies the crystallization dynamics and improves the film quality as evidenced by XRD (Supplementary Fig. 8) and SEM (Fig. 3a, b, d, e and Supplementary Table 3). Notably, the perovskite grains of films with *o*-CB showed perpendicular growth on the substrate (Fig. 3e), which may be attributed to the template-growth of grains assisted by *o*-CB, contributing to reduced grain boundaries (GBs), and more homogeneous and compact film morphology[29]. Peeling off the perovskite films using solidified glue (Supplementary Fig. 9), the buried surface of the pristine film showed nanovoids and nano-cracks at the bottom of the perovskite (Fig. 3c), while these were highly suppressed in the *o*-CB treated film (Fig. 3f), indicating an improved interfacial contact, which can reduce leakage current and improve device performance. The improved film quality also contributed to reduced Sn oxidation[28]. The lower Sn$^{4+}$ content in the film treated by *o*-CB than in the pristine film was confirmed by X-ray photoelectron spectroscopy (XPS) as shown in Supplementary Fig. 10.

We performed density functional theory (DFT) calculations to provide insight into interactions between *o*-CB and perovskite. The simulation results in Fig. 3g show that *o*-CB was adsorbed at the center of the four iodine atoms on the perovskite surface. The interactions between the two were dominated by van der Waals forces, with an adsorption energy of −0.32 eV, indicating that *o*-CB and perovskite can stably combine. This was supported by the charge density difference calculation, which showed that electrons of the iodine atom were attracted towards the *o*-CB side, resulting in a significant decrease in electron density in the upper layer of the perovskite (Fig. 3h, i), indicating the significant interactions between *o*-CB and perovskite. The electron transfer was further confirmed by the electronic location function (ELF)[30], which shows that the electron profile of iodine interacting directly with *o*-CB was slightly reduced (Supplementary Figs. 11 and 12). Bader charge analysis was implemented to quantitatively demonstrate electron transfer between the two. The analysis revealed a charge transfer of 0.3 e$^-$ from the perovskite to *o*-CB. The electron cloud density of *o*-CB is larger, making it easier for electrons to escape into a vacuum, thereby raising the fermi energy level, which was confirmed by VASP calculation (Supplementary Table 4).

The improved interfacial energy level was verified by the ultraviolet photoelectron spectroscopy (UPS) measurements (Supplementary Figs. 13 and 14). The valence band maximum of a perovskite film shifted from −5.42 to −5.31 eV after *o*-CB treatment, making its band offset with the highest occupied molecular orbital of PEDOT:PSS smaller and reducing interfacial recombination. It has also been noted that the Fermi level (E$_f$) of the *o*-CB-treated film upshifted 0.33 eV (Supplementary Fig. 15), suggesting a more n-type doped characteristic, which is in good agreement with the suppressed Sn oxidation as discussed above (Supplementary Fig. 10)[31]. The improved perovskite film quality and interfacial contact were further confirmed by the increased photoluminescence (PL, Supplementary Fig. 16) intensity and prolonged Time-resolved photoluminescence (TRPL) lifetime increasing from 358 ns for the pristine film to 630 ns for the films after *o*-CB treatment (Supplementary Fig. 17 and Supplementary Table 5). We further described the PL decay using the differential lifetime ($\tau_{PL}$),

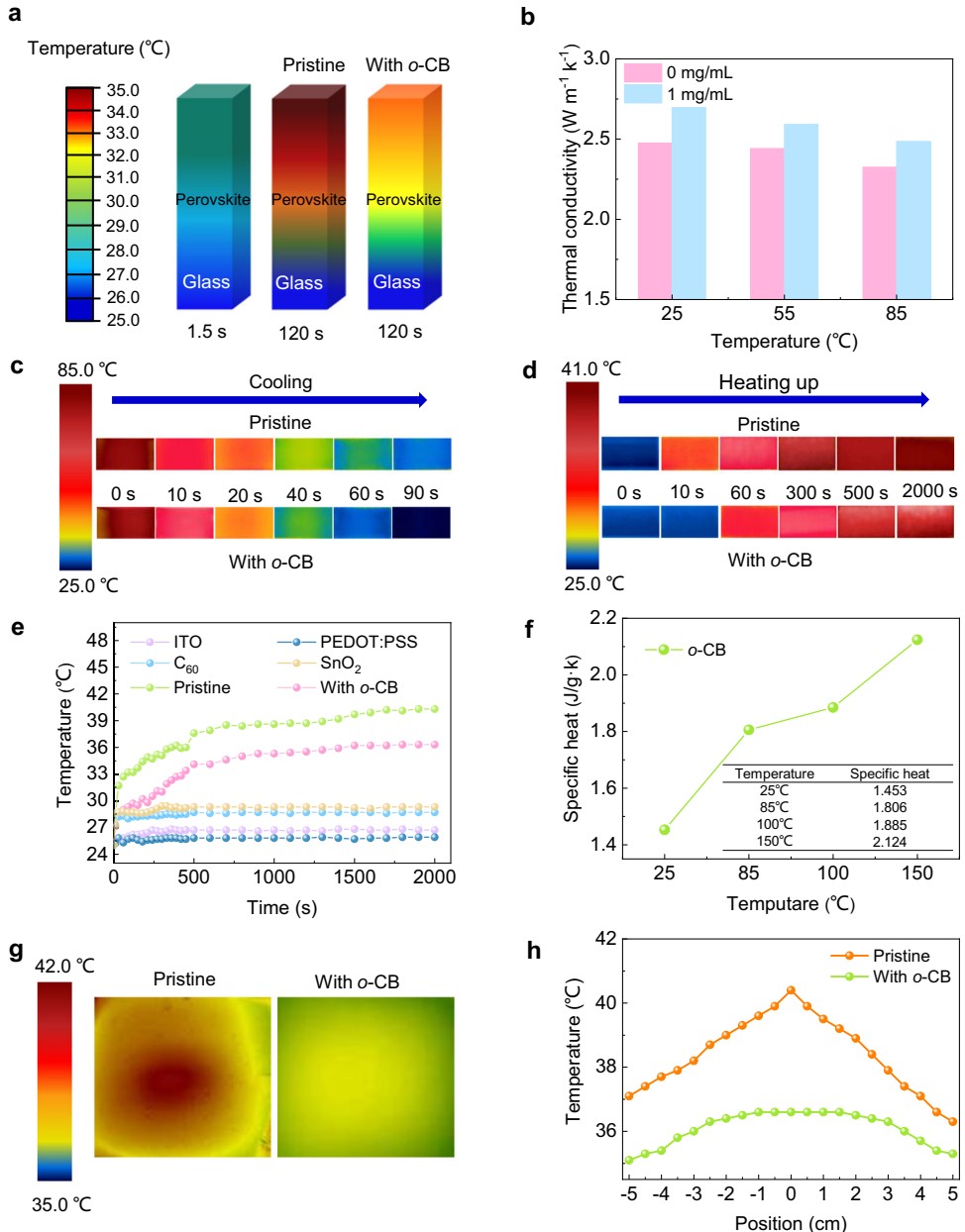

**Fig. 2 | The improvement of heat transfer. a** Software simulation analysis of the heat absorption for perovskite films treated with/without *o*-CB. The simulation within 1.5 s can accurately represent the same initial temperature point for both the pristine film and the film treated with *o*-CB. **b** Thermal conductivity of the perovskite films treated with different concentrations of *o*-CB. **c** Comparison of the cooling process of pristine perovskite film and perovskite film with *o*-CB through IR thermal images. The architecture of the testing films is ITO/PEDOT:PSS/perovskites w/ and w/o *o*-CB. **d** Comparison of the heating up process of pristine perovskite film and perovskite film with *o*-CB through IR thermal images. The architecture of the testing films is ITO/PEDOT:PSS/perovskites w/ and w/o *o*-CB. **e** Continuous recording of the temperature change of each layer of the device under illumination (AM 1.5 G, 100 mW cm⁻²). **f** Specific heat capacity change of the *o*-CB molecule at different temperatures. **g** IR thermal images of pristine perovskite film (5 cm * 5 cm) and perovskite film with *o*-CB treatment (5 cm * 5 cm) under 1-sun illumination for 2000 s. **h** Temperature distribution of perovskite films. The zero point of the *x*-axis refers to the center of the films.

which can be estimated in terms of the following Eq. (1):

$$\tau_{PL} = \left( -\frac{1}{m} \frac{d\ln(\Phi_{PL})}{dt} \right)^{-1} \tag{1}$$

where $\Phi_{PL}(t)$ is the PL intensity at $t$ after the photoexcitation, and m is a factor in relation to the injection level, which is set as 2 in this work. By solving the Equation, the results are shown in Supplementary Fig. 18 by plotting $\tau_{PL}$ as a function of the logarithm of PL intensity $\ln(\Phi_{PL})$. According to the platform part of the data result curve, we extracted the effective Shockley reading Hall (SRH) lifetime in the bulk (glass/

perovskite) of $\tau\frac{o-CB}{SRH} = 3.19$ µs and $\tau\frac{Ctrl}{SRH} = 278.6$ ns. Depending on the defect concentration, SRH lifetime is related to the defect-mediated recombination in perovskite bulk. It demonstrates that perovskite film containing *o*-CB has a higher SRH lifetime. The reduction of the SRH recombination center of the film with *o*-CB treatment ushered in a higher SRH lifetime, which also proved that the defect density of the film after *o*-CB treatment was lower.

## Solar cell efficiency and stability

To investigate the effect of *o*-CB on the photovoltaic performance, we fabricated inverted p-i-n PSCs with an ITO/PEDOT:PSS/perovskite/C₆₀/

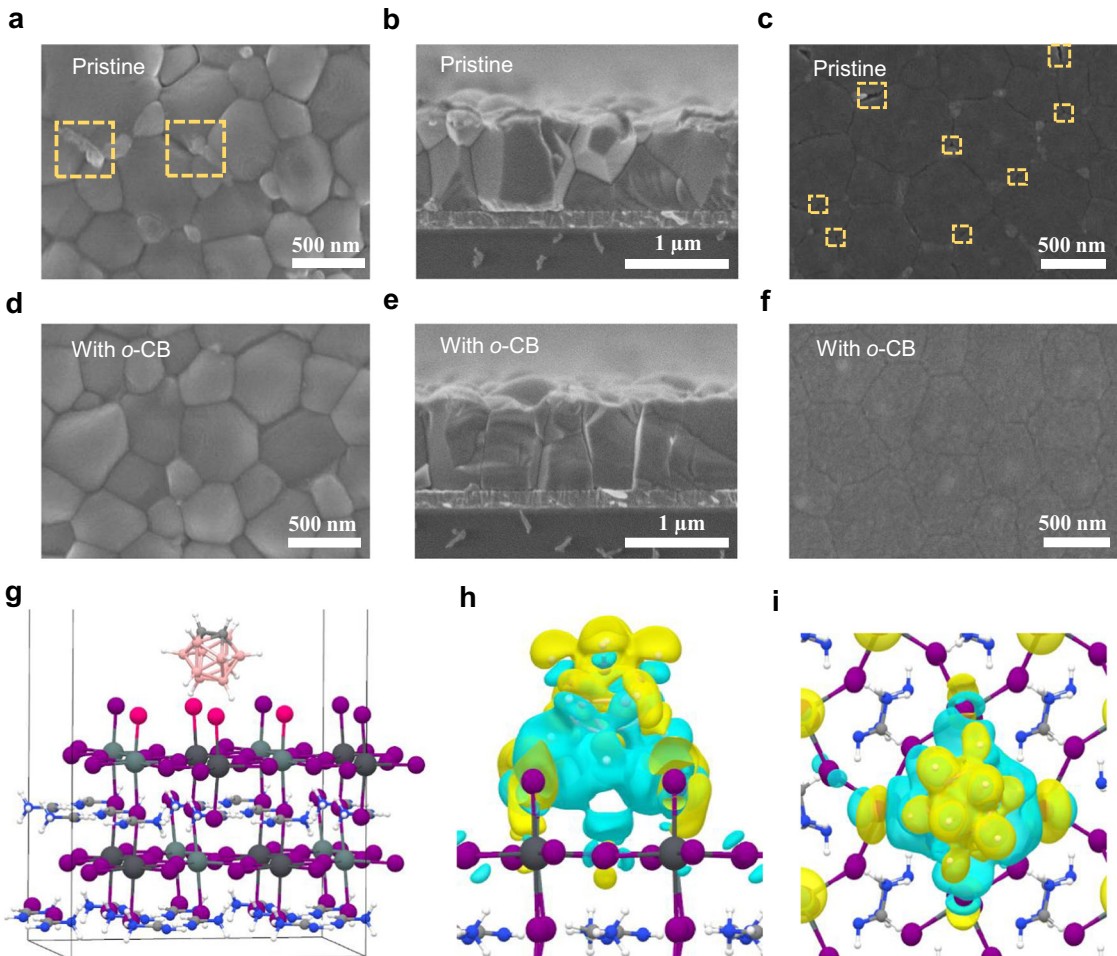

**Fig. 3 | Morphology characterization of perovskite films and DFT theoretical calculations.** Top-view SEM images of the surface of perovskite films **a** without and **d** with *o*-CB. Cross-sectional SEM images of perovskite films. The scale bar is 500 nm. **b** without and **e** with *o*-CB. The scale bar is 1 μm. Buried bottom surfaces of the perovskite films **c** without and **f** with *o*-CB. The scale bar is 500 nm. **g** Optimized structure of perovskite with *o*-CB, where the *o*-CB molecule is located at the center of the several highlighted iodine atoms. **h** Front and **i** top views of the charge density difference between *o*-CB and perovskite. The blue color represents a region of electron loss after binding, and the yellow color represents a region of electron aggregation.

SnO$_2$/Ag configuration. The optimal concentration of *o*-CB for obtaining the highest efficiency was 1.0 mg/mL (Supplementary Fig. 19). Transient photovoltage (TPV) decay measurements was first conducted to investigate the charge transport properties[32]. The TPV decay lifetime of the *o*-CB-treated device was 48 μs, compared with 34 μs for the pristine device (Fig. 4a), indicating enhanced hole extraction and reduced charge accumulation at the interface. The statistics of the performance parameters indicated an overall improvement in all parameters and better reproducibility after *o*-CB treatment (Supplementary Fig. 20 and Supplementary Table 6). The PCE of the champion device after *o*-CB treatment was 23.4%, with a short-circuit current density ($J_{SC}$) of 32.19 mA cm$^{-2}$, an open-circuit voltage ($V_{OC}$) of 0.877 V, and a fill factor (FF) of 0.829, surpassing that of the pristine one (Fig. 4b). In addition, the *o*-CB treated device exhibited lower hysteresis (Supplementary Fig. 21). At higher temperatures, charge accumulation occurs, causing higher charge recombination losses, thereby causing the occurrence of hysteresis. The device treated with *o*-CB can reduce charge accumulation under illumination conditions, decreasing charge recombination losses and thus suppressing the *J–V* hysteresis. Figure 4c shows the external quantum efficiency (EQE) spectra of champion devices w/ and w/o being treated with *o*-CB. The integrated current density of 31.0 mA cm$^{-2}$ and 30.2 mA cm$^{-2}$ agreed well with the values from *J–V* measurements. The stabilized power output (SPO) measured at the maximum power point (MPP) for devices w/ and w/o *o*-CB treatment resulted in a stabilized PCE of 22.94% and 20.18% without decay for 300 s, respectively (Fig. 4d). We tested the stability of mixed Sn-Pb solar cells at 85 °C. The *o*-CB treated devices retained 80% of the original PCE after 1080 h, while the pristine devices maintained only 40% of the initial PCE after ~200 h (Fig. 4e). The enhanced thermal stability might stem from the following reasons: (1) the improved thermal conductivity of *o*-CB, which suppresses thermal accumulation[33]; (2) the enhanced contact between perovskite and HTL and (3) a more appropriate band alignment at the buried interface[34], thereby reducing non-radiative recombination and the generation of phonons (heat). In addition, the introduction of *o*-CB also improves the crystallization of perovskite, which also contributes to enhanced thermal stability[35]. However, the improved crystallization of perovskite cannot invalidate the increase of thermal conductivity due to the introduction of *o*-CB that leads to improved thermal stability of the device (Supplementary Fig. 22). Figure 4f exhibits the MPP tracking of the mixed Sn-Pb solar cells w/ and w/o *o*-CB treatment under continuous one-sun illumination. The *o*-CB-treated device retained 90% of its initial PCE after 1000 h. In comparison, the pristine device retained only 62% of its original PCE after 400 h. The *o*-CB-treated device also showed better storage stability (Supplementary Fig. 23), retaining 96% of its initial PCE after 2000 h under an N$_2$ atmosphere. The universality of this approach was also verified by the significantly improved performance

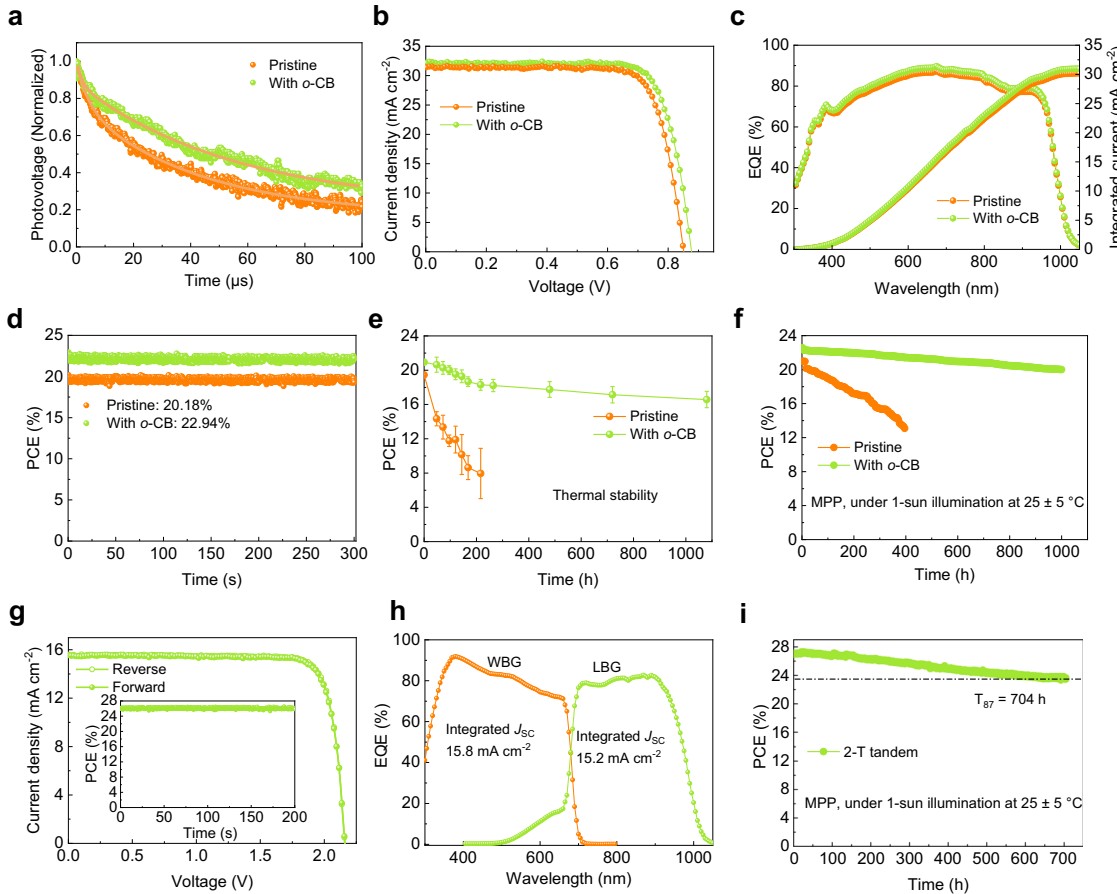

**Fig. 4 | Photovoltaic performance and stability. a** Normalized TPV of the devices w/ and w/o *o*-CB. **b** *J*–*V* characteristics of the champion devices w/ and w/o *o*-CB. **c** EQE spectra and integrated *J*$_{SC}$ of the devices w/ and w/o *o*-CB. **d** Stable efficiency output of the devices w/ and w/o *o*-CB at the maximum power point. **e** Thermal stability of the devices w/ and w/o *o*-CB at 85 °C. The error bars represent the standard deviation of 4 devices. **f** MPP tracking of the devices w/ and w/o treated by *o*-CB. **g** *J*–*V* curves of the best-performing tandem device under forward and reverse voltage scan. Inset, the SPO of the tandem device. **h** EQE spectra of the best-performing tandem device. **i** MPP tracking of the tandem device treated by *o*-CB.

and thermal stability of PSCs based on pure lead perovskite after introducing *o*-CB (Supplementary Figs. 24 and 25).

We integrated *o*-CB treated low-bandgap mixed Sn-Pb subcells into monolithically integrated all-perovskite tandem solar cells (Supplementary Fig. 26). The *J*–*V* curves of the tandem and wide-bandgap solar cells are shown in Fig. 4g and Supplementary Fig. 27. We achieved a champion PCE of 27.2 (27.1) % for the tandem solar cells, with *V*$_{OC}$ of 2.16 (2.15), *J*$_{SC}$ of 15.6 (15.7) mA/cm² and FF of 81 (80) % under reverse (forward) scan direction (Table 1). The SPO of the tandem solar cell was kept at 27% over 200 s (inserted in Fig. 4g). The current densities of the subcells calculated from EQE were 15.8 and 15.2 mA/cm² (Fig. 4h), consistent with the *J*–*V* measurements. We further tested the stability of the encapsulated tandem solar cells under simulated one-sun illumination at MPP. The pristine tandem retained 87% of its initial PCE

after 704 h of continuous operation (Fig. 4i). Further improving the stability of tandem solar cells lies in the enhancement of wide-bandgap subcells, such as the suppression of I-Br segregation under illumination as shown in Supplementary Fig. 28[36].

## Discussion

In this study, we have developed a sustainable strategy that utilizes electron-delocalized carboranes *o*-CB to regulate heat dissipation and facilitate charge transfer in mixed Sn-Pb and tandem solar cells. Through the application of *o*-CB treatment, the occurrence of localized areas with heat accumulation (hotspots) was efficiently suppressed, resulting in improved long-term thermal stability. Mixed Sn-Pb cell retained 80% of its initial PCE after 1080 h aging at 85 °C. *o*-CB treatment also reduced interface recombination. We achieved maximum PCEs of 23.4% and 27.2% for mixed Sn-Pb single-junction and all-perovskite tandem solar cells. The tandem remained at 87% of its initial PCE after 704 h of continuous operation under illumination.

## Methods
### Materials and solvents
All chemicals were purchased from commercial distributors and are used directly upon receipt without further processing. Methylammonium iodide (MAI, ≥99.5%), methylammonium bromide (MABr), bathocuproine (BCP, >99%), Phenethylammonium iodide (PEAI) and PEDOT:PSS (CLEVIOS P VP AI 4083 from Heraeus) were purchased from Xi'an Polymer Light Technology Corp. Formamidine iodide (FAI),

**Table 1 | PV parameters of narrow-bandgap, wide-bandgap PSCs and all-perovskite tandem solar cells**

| Device | | *J*$_{SC}$ (mA cm⁻²) | *V*$_{OC}$ (V) | FF (%) | PCE (%) |
|---|---|---|---|---|---|
| NBG | Reverse | 32.19 | 0.877 | 82.9 | 23.4 |
| | Forward | 32.02 | 0.874 | 83.1 | 23.26 |
| WBG | Reverse | 18 | 1.26 | 82.9 | 18.8 |
| | Forward | 17.95 | 1.28 | 80.9 | 18.59 |
| Tandem | Reverse | 15.6 | 2.16 | 81 | 27.2 |
| | Forward | 15.7 | 2.15 | 80 | 27.1 |

lead (II) iodide (PbI$_2$) methylamine and chloride (MACl) were purchased from Advanced Election Technology Co. Ltd. Tin (II) iodide (SnI$_2$, 99.999%), tin (II) fluorine (SnF$_2$, 99%), lead (II) bromide (PbBr$_2$, >98%), cesium iodide (CsI, >99.0%), *ortho*-Carborane (*o*-CB, >98.0%), [4-(3,6-Dimethyl-9H-carbazol-9-yl)butyl]phosphonic acid (Me-4PACz, >99.0%), were purchased from TCI. C$_{60}$ was purchased from Luminescence Technology Corp. Tetrakis(dimethylamino)tin(IV) (99.9999%) for atomic-layer-deposited (ALD) SnO$_2$ was bought from Nanjing Ai Mou Yuan Scientific Equipment. N,N-dimethylformamide (DMF, 99.8%, Acros), dimethyl sulfoxide (DMSO, 99.9%, Acros), isopropanol (IPA, 99.5%, Acros), ethyl acetate (EA, Acros), diethyl ether (Acros), chlorobenzene (CB, 99.8%, Acros) and methanol (Acros) were from Beijing Innochem Science & Technology co., LTD.

**Precursor solution preparation**
**The NBG mixed Sn-Pb Cs$_{0.1}$MA$_{0.2}$FA$_{0.7}$Pb$_{0.5}$Sn$_{0.5}$I$_3$ perovskite precursor**. The precursor solution (1.8 M) was prepared by dissolving PbI$_2$ (415 mg), SnI$_2$ (335.3 mg), FAI (216.8 mg), MAI (57.4 mg), CsI (46.8 mg), SnF$_2$ (9.4 mg), were dissolved in a mixture of DMSO: DMF (*v:v* = 3:1) solvent. Then 0 mg mL$^{-1}$、1 mg mL$^{-1}$ and 2 mg mL$^{-1}$ of additive *o*-CB were dissolved into the precursor mixture, and finally, the magnets were put into the mixed solution and stirred on a stirrer to dissolve completely.

**The WBG FA$_{0.8}$Cs$_{0.2}$Pb(I$_{0.7}$Br$_{0.3}$)$_3$ perovskite precursor**. The precursor solution was prepared by dissolving PbI$_2$ (233.2 mg), FAI (126.6 mg), CsI (47.8 mg), PbBr$_2$ (152.0 mg), were dissolved in a mixture of DMF: DMSO (*v:v* = 3:1) solvent.

**The Cs$_{0.05}$MA$_{0.05}$FA$_{0.9}$Pb(I$_{0.97}$Br$_{0.03}$)$_3$-based perovskite precursor solution**. The precursor solution was prepared by dissolving PbI$_2$ (689.2 mg), PbBr$_2$ (27.5 mg), FAI (232.8 mg), MABr (7.9 mg), CsI (19.4 mg), MACl (25.3 mg) dissolved in a mixture of DMF: DMSO (*v:v* = 4:1) solvent. Then 0 mg mL$^{-1}$, 0.5 mg mL$^{-1}$, and 1 mg mL$^{-1}$ of additive *o*-CB were dissolved into the precursor mixture.

**Device fabrication**
**Substrate cleaning.** The ITO glass substrate was placed in an alkaline cleaning solution, deionized water, acetone, and ethanol cleaning in turn, and then in the oven to dry after cleaning. Before using the ITO glass substrate, it should be treated with O$_2$ plasma cleaner for 600 s to make the surface hydrophilic.

**Device fabrication for NBG mixed Sn-Pb PSCs.** In this experiment, perovskite solar cells with an inverted p-i-n structure were assembled. Firstly, the HTL was prepared by taking 80 μL of PEDOT:PSS filtered with 0.22-μm PTFE and suspended on the ITO glass substrate at 4000 rpm for 30 s with an acceleration of 1500 r.p.m. s$^{-1}$, followed by annealing at 140 °C for 15 min. After annealing, the substrates were transferred into an N$_2$-filled glove box for the subsequent steps. Next, the precursor solution was filtered by 0.22-μm PTFE and coated on PEDOT:PSS at 1000 r.p.m. for 10 s, with an acceleration of 500 r.p.m. s$^{-1}$, then CB was dropped at a suitable speed in drops on the substrate at the 20th second before the end of the second step (4000 r.p.m. for 40 s). After suspension, the film was immediately transferred to a hot plate at 100 °C for 10 min. 25 nm of C$_{60}$ were evaporated. The substrates were then transferred to an ALD system to deposit SnO$_2$ (20 nm) at 85 °C. After that, the substrates were transferred to an evaporator to deposit 90 nm Ag.

**Device fabrication for all-perovskite tandem solar cells.** Me-4PACZ (1 mg mL$^{-1}$ dissolved in MeOH) was spin-coated onto the ITO. The hole transport layer Me-4PACZ (1 mg mL$^{-1}$) was coated on the ITO glass substrates at 3000 r.p.m. for 25 s, followed by 100 °C annealing for 10 min. Then DMF (70 μL) was spin-coated at 4000 r.p.m. for

10 s to improve the wettability of the Me-4PACZ surface prior to the WBG perovskite deposition. The perovskite precursor was spin-coated at 500 r.p.m. for 3 s and then at 4000 r.p.m. for 60 s. During the second step (4000 r.p.m. for 60 s), the diethyl ether (750 μL) was applied dropwise at a suitable speed in the center of the substrate to facilitate the crystallization. After that, the substrate was annealed at 60 °C for 2 min and at 100 °C for 5 min. 25 nm of C$_{60}$ was evaporated. The substrates were then transferred to an ALD system to deposit SnO$_2$ (20 nm) at 85 °C. After that, the substrates were transferred to an evaporator to deposit 1 nm Au. Then PEDOT:PSS was coated with Au spinning at 4000 r.p.m. for 30 s. After the end of spin coating, it was heated and annealed at 100°C for 10 min. After cooling, the substrates were transferred to the N$_2$-filled glove box to deposit the NBG subcells using the above-mentioned method. Firstly, the HTL was prepared by taking 80 μL of PEDOT:PSS filtered with 0.22-μm PTFE and suspended on the substrates at 4000 rpm for 30 s with an acceleration of 1500 r.p.m. s$^{-1}$, followed by annealing at 140 °C for 15 min. After annealing, the substrates were transferred into an N$_2$-filled glove box for the subsequent steps. Next, the precursor solution was coated on PEDOT:PSS at 1000 r.p.m. for 10 s, with an acceleration of 500 r.p.m. s$^{-1}$, then CB was dropped at a suitable speed in drops on the substrate at the 20th second before the end of the second step (4000 r.p.m. for 40 s). After suspension, the film was immediately transferred to a hot plate at 100 °C for 10 min. 25 nm of C$_{60}$ were evaporated. The substrates were then transferred to an ALD system to deposit SnO$_2$ (20 nm) at 85 °C. After that, the substrates were transferred to an evaporator to deposit 90 nm Ag.

**Device fabrication for Cs$_{0.05}$MA$_{0.05}$FA$_{0.9}$Pb(I$_{0.97}$Br$_{0.03}$)$_3$-based.** The device fabrication was carried out in a glove box. First, the HTL Me-4PACZ was prepared at a concentration of 1 mg mL$^{-1}$. The prepared solution was spin-coated onto the ITO substrate at 3000 r.p.m. for 25 s and then annealed at 100 °C for 10 min. After cooling, the precursor solution filtered through 0.22-μm PTFE was coated on the top of HTL at 2500 r.p.m. for 10 s, followed by 5000 r.p.m. for 40 s, and the anti-solvent EA was slowly dripped at the 30th second before the end of the second step (5000 r.p.m. for 40 s). After suspension coating, the films were immediately transferred to a hot plate at 100 °C for 30 min. After cooling, PEAI (1 mg mL$^{-1}$ dissolved in IPA) was coated to the perovskite layer as a passivation layer at 5000 r.p.m. for 25 s and annealed at 100 °C for 10 min. After cooling, the layer was cleaned with IPA. 25 nm of C$_{60}$ was evaporated. The substrates were then transferred to an ALD system to deposit SnO$_2$ (20 nm) at 85 °C. After that, the substrates were transferred to an evaporator to deposit 90 nm Ag.

**Characterization**
**Thermal characterization.** The instrument used to test the thermal conductivity was the Hotdisk model DRL-III. The temperature of each layer was measured by a T thermocouple, model YET-610, 1 channel, at a measurement rate of 2 times s$^{-1}$. The thermal images were obtained by using a thermal infrared camera (FLIR T640).

**Ansys icepak.** The Ansys Icepak was used to simulate the device's heat dissipation process. The physical model of the ITO glass layer and perovskite layer are $0.17 \times 0.18 \times 0.01 \, mm^3$ and $0.17 \times 0.18 \times 0.0005 \, mm^3$. The mesh dissection was performed to partition the model into computable mesh models. The initial temperature was set to 25 °C. The heat dissipation process between the perovskite and the contact surface was heat conduction transfer, and the heat dissipation process between the perovskite and the environment was convection heat transfer. The period of the whole heat transfer process was 10 s, and no boundary conditions were set for the whole process.

**Time of Flight Secondary Ion Mass Spectrometry (TOF-SIMS).** TOF-SIMS test equipment model was ION TOF-SIMS 5 from Ion-Tof GmbH, Germany.

**X-ray diffraction (XRD).** XRD was measured by Cu Kα (1.5406 Å) radiation, using an Ultima IV type ray diffractometer.

**Scanning electron microscopy (SEM).** SEM measurements were performed using a field emission scanning electron microscope (Hitachi, S4800).

**X-ray photoelectron spectroscopy (XPS).** XPS spectra were performed for films on ITO using a Thermo Fisher ESCALAB 250 Xi under air. Curve fitting was performed using the Thermo Avantage software. The curves were corrected based on the C1$s$ peak at 284.8 eV.

**Ultraviolet photoelectron spectroscopy (UPS).** The UPS (AXIS ULTRA DLD, Kratos, UK) equipped with a He-I source ($hv = 21.22$ eV) was used to measure the conduction band, valence band energies, and Fermi energy levels of the thin film samples.

**Ultraviolet–visible (UV–Vis).** The UV–Vis absorbance was measured by UV/Vis spectrometer (Ocean Optics).

**Photoluminescence (PL).** Steady-state PL measurements were performed using an Edinburgh F900 spectrometer with an excitation wavelength of 500 nm to excite the sample from the perovskite film side. The PL signal was then detected with a Symphony-II CCD (charge-coupled device; Horiba) detector.

**Time-resolved photoluminescence (TRPL).** The time-resolved PL was tested using an Edinburgh F980 spectrometer with an excitation wavelength of 600 nm using a single photon counting technique.

**Transient photovoltage (TPV).** TPV was measured by a self-made laser pulse oscilloscope.

**PSC characterization.** The $J–V$ characteristics of the devices (voltage scanning rate 10 mV/30 ms) and the steady photocurrent under maximum power output bias were recorded under AM 1.5 G illumination at 100 mW cm$^{-2}$ with a solar simulator (PV Measurements Inc.) under ambient conditions. $J–V$ curves for all devices were measured by masking the devices with a metal mask with an aperture area of 0.09 cm$^2$. The light intensity of the solar simulator was calibrated with a standard silicon solar cell certified by the National Renewable Energy Laboratory. The EQE spectra (300–1100 nm) were obtained by QE Systems (PV Measurements Inc.). All the measurements were carried out under an ambient atmosphere.

**Thermal stability test.** All unencapsulated PSCs were tested by placing the complete devices on a heating plate at 85 °C in a glove box. All the devices are cooled down before each test in ambient air with a relative humidity of ~ 30% within 5 min.

**Operational stability test.** All unencapsulated PSCs were tested for MPP stability using the stability setting (LC Auto-Test 24, Shenzhen Lancheng Technology Co., Ltd.). All PSCs were tested under continuous illumination and maximum power point. During the aging process, the light source consisted of an array of white LEDs (SLS-LED-80, Qingdao Solar Scientific Instrument High-tech Co., LTD) powered by a constant current. Sun intensities were calibrated by a calibrated Si-reference cell. During aging, the devices were masked and placed in a holder purged with continuous N$_2$ flow.

**N$_2$ stability test.** All devices were unencapsulated and placed in a glove box under an N$_2$ atmosphere and treated with tinfoil for light-proof storage. The devices were transferred to an air environment with a relative humidity of approximately 30% and at room temperature for each test.

## Reporting summary

Further information on research design is available in the Nature Portfolio Reporting Summary linked to this article.

## Data availability

The data that support the plots within this paper and other findings of this study are available from the corresponding author upon request.

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

## Acknowledgements
This work was supported by the Taishan Scholars Project of Shandong Province (Grant No: 20190921) and the National Natural Science Foundation of China (22109076). The authors also acknowledge Prof. Maning Liu from Lund University for TRPL analysis in this research.

## Author contributions
Z.Z. directed and supervised the project. S.T., C.L. and Z.Z. proposed the idea and designed the experiment. S.T., C.L. and C.P. fabricated and characterized the PSCs. W.Y. and S.T. conducted TPV, XPS, PL, and XRD measurements. C.P. conducted SEM measurements of buried surfaces. H.B. and H.G. performed DFT calculations. H.J. performed the specific heat calculations. L.Z. provided the carboranes. F. Y. conducted data analysis. The manuscript was prepared, revised, and finalized by S.T., C.L. and Z.Z. All authors discussed the results and commented on the manuscript.

## Competing interests
The authors declare no competing interests.
