## [Peer Review File · Nature Communications]

Sustainable thermal regulation improves stability and efficiency in all-perovskite tandem solar cellsREVIEWER COMMENTS

Reviewer #1 (Remarks to the Author):

Improving the thermal and operation stability of Pb-Sn perovskite solar cells is critical for the commercialization of all-perovskite tandem solar cells. This manuscript takes the heat transfer model as the starting points, providing us a new perspective to evaluate the performance and application of Pb-Sn perovskite solar cells.

The authors discovered the regulation of heat dissipation and charge transfer by introducing o-CB into Pb-Sn perovskite precursor solution. Through the o-CB treatment, the occurrence of localized areas with heat accumulation (hotspots) was efficiently suppressed, resulting in improved long-term thermal stability. Mixed Pb-Sn sub-cells retained 80% of its initial PCE after 1,080 h aging at 85 °C. The o-CB treatment also reduced interface recombination. They achieved maximal in-lab measured PCEs of 23.44% and 27.2% for mixed Pb-Sn single junction and all-perovskite tandem solar cells. The tandem remained at 87% of its initial PCE after 704 h of continuous operation under illumination.

However, there are several crucial scientific problems before considering its suitability for publication in Nat Comm, especially in heat transfer model and performance measurement of tandem cells. Meanwhile, there is a lack of convincing characterization to demonstrate the effects of this additive. The detailed comments are as follows:

1. In Fig. 2c and 2d, perovskite films with o-CB show a faster cooling and slower heating up. How can we explain that perovskite films with o-CB exhibit a thermal diode-like effect? Moreover, the schematic diagram in Fig.1b shows that o-CB is deposited at the bottom of the perovskite film, which can be understood as an ultra-thin conductor in series between PEDOT: PSS and perovskite, so how does such a small amount of materials affect the thermal conductivity change of the entire device? The authors need to show the thermal conductivity change of the whole device to analyze this problem, not the change of the sole perovskite film.
2. In Fig.2g, the o-CB perovskite films show better thermal conductivity. So, if the target devices are placed into a hot environment (i.e., outdoor in summer), would the high temperature in the environment also be introduced into the target devices faster, resulting in poorer stability?
3. If mixed Pb-Sn perovskite precursor solution is 500 μL in the methods of this paper, we get the concentration is 2.4 M. However, SEM images in Fig. 3b and 3e exhibit an unreasonable thickness (~ 500 nm) for both the pristine and the o-CB. At this absorber-layer thickness, the EQE of Pb-Sn perovskite

solar cells at the long-wavelength would not be higher than 65% (<https://doi.org/10.1038/s41467-019-12513-x>), let alone 80% obtained in Fig.4c.

4. What is the substrate in TRPL shown in Supplementary Fig. 15, ITO/PEDOT: PSS or pure glass? It seems that the TRPL of the control films show an abnormal downward slope over a quite short period of time (<50 ns), while the o-CB films do not. In my opinion, a sudden change in early time could be down to charge transfer rather than recombination, thus the substrates for these two samples are critical. The authors need to explain it and further discuss the results of TRPL by relevant experimental or mathematical methods.

5. In Fig. 4c, the EQE curves of the control and o-CB devices show different initial EQE values. Does this mean that the improvement of Jsc comes from the anti-reflection layer?

6. In Fig.4e, the o-CB and control devices demonstrate 1080 h and 80 h for T80 at 85 oC, respectively, which is similar or even inferior compared with previous report also from the authors group (BBMS as additive, 2660 h of T98 at 65 oC). Why does not this work continue previous recipe, using BBMS as the control device and presenting better device thermal stability?

7. In all-perovskite tandem solar cells (Table 1), the o-CB tandem device shows a Voc of 2.16 V. However, LBG and WBG device exhibits a Voc of 0.88 V and 1.26 V, respectively. Could the authors further discuss how to minimize the Voc loss of the all-perovskite tandem solar cells? By the way, the o-CB tandem devices show a Jsc of 15.6 mA cm⁻², which are not agreement with the EQE results shown in Fig. 4h.

8. The manuscript stated “this approach was also verified by the significantly improved performance and thermal stability of PSCs based on pure lead perovskite after introducing o-CB” (line 218), and “the simulation results in Fig. 3g show that o-CB was adsorbed at the center of the four iodine atoms on the perovskite surface”(line 163). So, could o-CB be utilized in WBG sub-cells to inhibit I-Br segregation and achieve better operation stability for all-perovskite tandem solar cells?

9. The J-V characterization of tandem solar cells is not rigorous. Please refer to the IEC standards on how to correctly measurement the JV and EQE of tandem solar cells. An independent certification will be highly appreciated.

Reviewer #2 (Remarks to the Author):

The paper by Tan et al. aims to improve the stability of PbSn-based perovskite with a low bandgap of 1.25 eV, which are currently used for all-perovskite tandem cells, yet still face critical challenges with regard to their stability. This is generally understood to be due to the oxidation of Sn²⁺ to Sn⁴⁺. However, the authors identify that thermal conductivity plays a major role in their stability as insufficient thermal transfer can lead to heat accumulation within the absorber layer that accelerates thermal degradation. To mitigate heat accumulation, the authors incorporated carboranes into perovskite due to their heat transfer capability. Finally, high-performance tandem cells with efficiencies over 27% are demonstrated with enhanced stability under maximum power point tracking. The paper is well-written and structured and provides valuable insights into timely research questions. It also demonstrates highly efficient tandem cells. As such the paper may be acceptable for Nature Communications, however, further clarifications are necessary before the paper can be considered again.

While the argument of improved heat transfer is well presented with simulations and experiments, I am a bit surprised about the huge impact of a 1 mg/mL solution additive on the heat dynamics. First of all, an exceptional heat transfer ability is claimed in Figure S1. However, this claim lacks context, what is exceptional? To me the difference between PEDOT:PSS and o-CB in Figure S1 appears to be rather small. Please explain and provide a rationale for how this can significantly alter the temperature of the cell (e.g. as shown in Figure 2h). Does Figure 2h hold true for longer illumination, i.e. for operational conditions? Moreover, while the heat transfer ability has likely an impact on the stability, it is not really proven that this is the main factor for improved device stability. Can the authors prove, that a few degree differences in the temperature of the cell, let's say 4-5 °C (as shown in Figure 2h) can make such a big difference in the stability?

Other smaller points to consider for the authors.

How thick were the Films in Figure S1? Related to this how thick is the o-CB layer in the device (I guess this can be inferred from Fig. S2)? In Figures 1e and f, SEM images are provided of annealed samples at 85 degrees, please provide in addition original SEM images, this would strengthen the argument. Regarding Figure 2c and d, please provide in the caption or image the device structure (currently only provided in the main text). Regarding the statement "We then used T-type thermocouples to conduct real-time temperature monitor to investigate the photothermal response of each layer in PSCs" how can this work to detect the temperature of nm thick films? Please explain. Regarding the importance of electron transfer to o-CB, the authors may find a recent study interesting that demonstrated o-CB as an interfacial layer at the n-type interface which improved the electron transfer as well (<https://doi.org/10.1038/s41467-022-34203-x>). Regarding the energy levels shown in Figure S13, the electron affinity of C60 is likely not at 4.5 eV, this would cause too much recombination losses. Related to this, the argument that the energy level offset between PEDOT:PSS and the perovskite gets smaller when the energy level of the perovskite with the o-CB shift upwards, this argument would also imply that the C60 energy offset gets larger (with current energy levels), please reconsider this statement.

Reviewer #3 (Remarks to the Author):

This manuscript presents a strategy of introducing ortho-carborane (o-CB) to improve both efficiency and stability of mixed Sn-Pb perovskite solar cells and all-perovskite tandem solar cells. The incorporation of o-CB effectively transfers heat and lowers the surface temperature of perovskite film by around 5 °C under 1 sun illumination. As a result, the mixed Sn-Pb perovskite solar cell with o-CB treatment exhibits improved thermal stability, with 80% of its initial efficiency kept after aging at 85 °C for 1080 hours. Additionally, the introduction of o-CB improves the film quality and facilitates the charge transfer, enabling the remarkable efficiencies of 23.44% and 27.2% for mixed Sn-Pb cells and all-perovskite tandem cells, respectively. This work is of great interest and significance to the community of perovskite materials and solar cells, especially in mixed Sn-Pb cells and all-perovskite tandem cells, representing a significant advance. Therefore, I would recommend publication of this work after addressing the following minor concerns issues/comments.

1. Could the authors explain why they considered to use the compound of carboranes? Compared to other compounds, ortho-carbon borane has a higher specific heat capacity, which is beneficial for improving stability. In terms of device performance, have the other two compounds been tried, and what were the possible results?
2. "In comparison to the control film which exhibited a localized area of heat accumulation (hotspot), the o-CB treated film showed improved heat dissipation and a more uniform temperature distribution." Can the authors explain why the control film exhibits thermal accumulation in the middle?
3. "A heat dissipation model (Fig. 2a) with an initial temperature of 25°C and a size of 0.17 cm × 0.15 cm × 0.016 cm was established (see detailed model information 88 in the supplementary materials)." The authors' choice of 1.5 s instead of 0 s as a reference point should be clarified. It would be helpful if the authors could explain the rationale behind this selection. Additionally, it would be interesting to know if the inclusion of 1.5 s has any impact on the overall changes observed in the study.
4. The introduction of o-CB leads to significant reduction of the JV curve hysteresis of devices. The authors should discuss more to further elaborate on this.
5. The authors are suggested to provide more explanations on the statement of "The improved film quality also contributed to reduced Sn oxidation".
6. The authors provide the operational stability of the tandem cells. It is recommended that the authors also analyze and discuss why the thermal stability of the tandem cells gets improved.

Responses to reviewer comments

We thank the reviewers for their advice, which has helped us to improve the quality of the work. Below are responses to reviewers' questions. Author actions in response are in blue font below. Changes to the manuscript have been made in the tracking mode.

Reviewer #1 (Remarks to the Author):

Improving the thermal and operation stability of Pb-Sn perovskite solar cells is critical for the commercialization of all-perovskite tandem solar cells. This manuscript takes the heat transfer model as the starting points, providing us a new perspective to evaluate the performance and application of Pb-Sn perovskite solar cells. The authors discovered the regulation of heat dissipation and charge transfer by introducing o-CB into Pb-Sn perovskite precursor solution. Through the o-CB treatment, the occurrence of localized areas with heat accumulation (hotspots) was efficiently suppressed, resulting in improved long-term thermal stability. Mixed Pb-Sn sub-cells retained 80% of its initial PCE after 1,080 h aging at 85 °C. The o-CB treatment also reduced interface recombination. They achieved maximal in-lab measured PCEs of 23.44% and 27.2% for mixed Pb-Sn single junction and all-perovskite tandem solar cells. The tandem remained

at 87% of its initial PCE after 704 h of continuous operation under illumination. However, there are several crucial scientific problems before considering its suitability for publication in Nat Comm, especially in heat transfer model and performance measurement of tandem cells. Meanwhile, there is a lack of convincing characterization to demonstrate the effects of this additive. The detailed comments are as follows:

1. In Fig. 2c and 2d, perovskite films with *o*-CB show a faster cooling and slower heating up. How can we explain that perovskite films with *o*-CB exhibit a thermal diode-like effect?

Response: The exhibition of the diode-like effect is caused by the thermal conductivity difference in perovskites and *o*-CB. The faster cooling and slower heating up in our samples are attributed to the higher specific heat capacity and lower thermal conductivity of *o*-CB compared to perovskites. Specifically, during heating, *o*-CB with a high specific heat capacity will show a decrease in thermal conductivity along with the increase in temperature. Such characteristics will result in slower heat absorption and temperature rising, and thus cause faster cooling and slower heating up in our samples.

Moreover, the schematic diagram in Fig.1b shows that *o*-CB is deposited

at the bottom of the perovskite film, which can be understood as an ultra-thin conductor in series between PEDOT: PSS and perovskite, so how does such a small amount of materials affect the thermal conductivity change of the entire device?

Response: We here attribute the excellent heat dispersion ability of *o*-CB to its excellent intrinsic thermal conductivity. *o*-CB involves three-dimensional networks of atoms that allow efficient transmission of thermal energy through the material. It is also an electron-delocalized molecule with significant overlap between adjacent atomic orbitals. This allows electrons to move freely, facilitating heat transfer in the entire device. Considering that the perovskite film thickness is less than 1 μm , such an amount of *o*-CB should be enough for efficient heat transfer. This is seen in other literature (DOI: 10.1002/adfm.202308036; DOI: 10.1002/sml.202207092), where even less concentrated molecules were used to regulate the heat dispersion.

The authors need to show the thermal conductivity change of the whole device to analyze this problem, not the change of the sole perovskite film.

Response: We returned to the lab and could provide a thermal conductivity study of the complete devices (Fig. R1). The thermal conductivities of the

control devices were $1.443 \text{ W m}^{-1} \text{ k}^{-1}$, $1.42 \text{ W m}^{-1} \text{ k}^{-1}$, and $1.355 \text{ W m}^{-1} \text{ k}^{-1}$ at $25 \text{ }^\circ\text{C}$, $55 \text{ }^\circ\text{C}$, and $85 \text{ }^\circ\text{C}$. In contrast, the thermal conductivities of devices after *o*-CB treatment increased to $1.478 \text{ W m}^{-1} \text{ k}^{-1}$, $1.435 \text{ W m}^{-1} \text{ k}^{-1}$, and $1.362 \text{ W m}^{-1} \text{ k}^{-1}$ at $25 \text{ }^\circ\text{C}$, $55 \text{ }^\circ\text{C}$, and $85 \text{ }^\circ\text{C}$. The improvement in thermal conductivity after *o*-CB treatment is beneficial for the heat dissipation in whole devices. Fig. R1 has been added as Supplementary Fig. 4 to the revised Supporting Information.

Fig. R1. Thermal conductivities of the complete devices with perovskite films treated with/without *o*-CB.

2. In Fig.2g, the *o*-CB perovskite films show better thermal conductivity. So, if the target devices are placed into a hot environment (i.e., outdoor in summer), would the high temperature in the environment also be introduced into the target devices faster, resulting poorer stability?

Response: The high temperature in the environment will be introduced into

the target device faster because of the better thermal conductivity of *o*-CB, however, this will not cause poor stability. After putting solar cells into a hot environment, heat exchange happens, and shortly following will be the reaching of a thermal equilibrium state. Thus, it is the temperature when reaching thermal equilibrium that decides the stability, rather than the temperature rising rate. For a control device, the heat will accumulate at the interface after the device absorbs light, and more energy will be accumulated at the buried interface between the hole transport layer and the perovskite layer. In a target solar cell, the device with the introduction of *o*-CB can transfer heat more quickly, thus reducing heat accumulation and improving the stability of the device.

3. If mixed Pb-Sn perovskite precursor solution is 500 μ L in the methods of this paper, we get the concentration is 2.4 M. However, SEM images in Fig. 3b and 3e exhibit an unreasonable thickness (\sim 500 nm) for both the pristine and the *o*-CB. At this absorber-layer thickness, the EQE of Pb-Sn perovskite solar cells at the long-wavelength would not be higher than 65% (<https://doi.org/10.1038/s41467-019-12513-x>), let alone 80% obtained in Fig.4c.

Response: Thanks for reviewing our manuscript carefully. We returned to the lab and re-measured the SEM (Fig. R2). The control samples exhibit an

average thickness of 966 nm, with a standard deviation of ± 33 nm, and the target samples display an average thickness of 976 nm, with a standard deviation of ± 15 nm. We have updated the images and the thickness values in the revised manuscript. The thin films reported in our original manuscript are likely attributed to sample drift during the SEM measurements, which unintentionally caused compression of the measured images.

Fig. R2. Cross-sectional SEM images of perovskite films (a) without and (b) with *o*-CB.

4. What is the substrate in TRPL shown in Supplementary Fig. 15, ITO/PEDOT: PSS or pure glass? It seems that the TRPL of the control films show an abnormal downward slope over a quite short period of time (< 50 ns), while the *o*-CB films do not. In my opinion, a sudden change in early time could be down to charge transfer rather than recombination, thus the substrates for these two samples are critical. The authors need to explain it and further discuss the results of TRPL by relevant experimental or mathematical methods.

Response: Thank you for reviewing our manuscript carefully. The substrate used for TRPL measurements is glass. Since the charge transport layer is absent, we speculate the downward slope over a quite short time should be attributed to the bulk and surface non-radiative recombination in the perovskite thin films (10.1002/aenm.202003489).

5. In Fig. 4c, the EQE curves of the control and o-CB devices show different initial EQE values. Does this mean that the improvement of J_{sc} comes from the anti-reflection layer?

Response: We have not used any anti-reflection layer for our solar cells. We returned to the lab and fabricated more batches of solar cells to confirm the EQE characteristic. We found that although all the ITO glasses have the same sheet resistance, ITO glasses from different vendors provided us with different EQE characteristics, especially at the short wavelength range. This is likely caused by the difference in the quality of ITO from different vendors. We now used the same ITO glasses for solar cells reported in our manuscript, and the new $J-V$ and EQE data (Fig. R3) have been updated in the revised manuscript. These results are in good agreement with those reported originally.

Fig. R3. (a) J - V characteristics of the champion devices w/ and w/o *o*-CB. (b) EQE spectra and integrated J_{sc} of the devices w/ and w/o *o*-CB.

6. In Fig.4e, the *o*-CB and control devices demonstrate 1080 h and 80 h for T₈₀ at 85 °C, respectively, which is similar or even inferior compared with previous report also from the authors group (BBMS as additive, 2660 h of T₉₈ at 65 °C). Why does not this work continue previous recipe, using BBMS as the control device and presenting better device thermal stability?

Response: Thank you for the reviewer's attention to our previous work (DOI: 10.1021/acseenergylett.2c02576). The introduction of BBMS indeed improves the stability of the devices. As reported in our previous work, there was almost no efficiency decay when the devices were heated at 65°C for 2660 hours. However, when the heating temperature increased to 85°C, the stability significantly decreased for the device with BBMS. We investigated the thermal stability of the devices with BBMS at 85°C, but the results were not ideal as shown in Fig. R4. Although the devices with BBMS performed better than the control device under 85°C, the efficiency

decayed to 50% of its initial efficiency after 1000 hours. We speculate that the accelerated decay is attributed to the instability of the BBMS at high temperatures. In this work, we conducted more stringent tests at 85°C to investigate the influence of *o*-CB on the stability of the devices without using the BBMS recipe.

Fig. R4. Thermal stability of PSCs with and without BBMS annealing at 85 °C in an N₂ glovebox.

7. In all-perovskite tandem solar cells (Table 1), the *o*-CB tandem device shows a V_{OC} of 2.16 V. However, LBG and WBG device exhibits a V_{OC} of 0.88 V and 1.26 V, respectively. Could the authors further discuss how to minimize the V_{OC} loss of the all-perovskite tandem solar cells?

Response: In general, an all-perovskite tandem solar cell is composed of a narrow-bandgap sub-cell, a wide-bandgap sub-cell, and an interconnecting layer. The loss of open-circuit voltage (V_{OC}) primarily stems from these

three components. Therefore, minimizing the loss of V_{OC} requires: 1) reducing the defects in narrow-bandgap sub-cells, which originate from low-quality perovskite films and p-doping defects caused by the oxidation of Sn^{2+} to Sn^{4+} . 2) suppressing halide segregation in wide-bandgap perovskite, which is considered a major source of V_{OC} losses in wide-bandgap devices. 3) improving the quality of the interconnecting layer to reduce series resistance and eliminate lateral shunt pathways.

In this work, we specifically paid attention to the optimization of the interconnection layer. We meticulously optimized both the ALD- SnO_2 thickness and the ALD annealing temperature. Our findings indicate that depositing 20 nm SnO_2 at a temperature of 85°C allows us to assemble tandem solar cells with negligible V_{OC} loss. The ALD annealing temperature has been added to the Method section in the revised manuscript.

By the way, the *o*-CB tandem devices show a J_{SC} of 15.6 mA cm^{-2} , which are not agreement with the EQE results shown in Fig. 4h.

Response: Thank you for reviewing the manuscript carefully. We checked that the integrated current density is 15.2 mA cm^{-2} , which should fall within a reasonable margin of error compared to the 15.6 mA cm^{-2} obtained from the device.

8. The manuscript stated “this approach was also verified by the significantly improved performance and thermal stability of PSCs based on pure lead perovskite after introducing *o*-CB” (line 218), and “the simulation results in Fig. 3g show that *o*-CB was adsorbed at the center of the four iodine atoms on the perovskite surface”(line 163). So, could *o*-CB be utilized in WBG sub-cells to inhibit I-Br segregation and achieve better operation stability for all-perovskite tandem solar cells?

Response: In the original manuscript, we did not investigate the influence of *o*-CB on wide-bandgap sub-cells, but rather studied its effect on normal-bandgap devices. According to the suggestion from the reviewer, we introduced *o*-CB into the wide-bandgap perovskite films and found that *o*-CB can suppress I-Br separation, as shown in Fig. R5. Therefore, the high operational stability of the all-perovskite tandem solar cell is attributed to the stable narrow-bandgap sub-cell and wide-bandgap sub-cell due to the introduction of *o*-CB. This reason has been added to the revised manuscript and Fig. R5 has been added as Supplementary Fig. 26 to the revised Supporting Information.

Fig. R5. XRD patterns of the perovskite films a) without and b) with *o*-CB treatment before and after 100 mW cm^{-2} white light illumination for 2 days.

9. The J - V characterization of tandem solar cells is not rigorous. Please refer to the IEC standards on how to correctly measurement the JV and EQE of tandem solar cells.

Response: Thank you for your suggestion. We have carefully read the IEC standards on how to correctly measure the J - V and EQE of tandem solar cells, and our rather outdated testing equipment may have resulted in less rigorous curves.

An independent certification will be highly appreciated.

Response: We appreciate the suggestion from the reviewer. However, as our research primarily centers on exploring novel materials for thermal regulation to improve solar cell stability, an independent efficiency

certification falls slightly beyond the scope of our work.

Reviewer #2 (Remarks to the Author):

The paper by Tan et al. aims to improve the stability of PbSn-based perovskite with a low bandgap of 1.25 eV, which are currently used for all-perovskite tandem cells, yet still face critical challenges with regard to their stability. This is generally understood to be due to the oxidation of Sn²⁺ to Sn⁴⁺. However, the authors identify that thermal conductivity plays a major role in their stability as insufficient thermal transfer can lead to heat accumulation within the absorber layer that accelerates thermal degradation. To mitigate heat accumulation, the authors incorporated carboranes into perovskite due to their heat transfer capability. Finally, high-performance tandem cells with efficiencies over 27% are demonstrated with enhanced stability under maximum power point tracking. The paper is well-written and structured and provides valuable insights into timely research questions. It also demonstrates highly efficient tandem cells. As such the paper may be acceptable for Nature Communications, however, further clarifications are necessary before the paper can be considered again.

While the argument of improved heat transfer is well presented with simulations and experiments, I am a bit surprised about the huge impact of a 1 mg/mL solution additive on the heat dynamics.

Response: We thank the reviewer for the careful evaluation of the manuscript. We fully understand the concern raised by the reviewer. We speculate that there are two main reasons for this:

1) Heat accumulation at the interface between the hole transport layer and the perovskite layer can be mitigated by *o*-CB, which can facilitate faster heat transfer and reduce the temperature at the buried interface. There are other papers reporting that even smaller amounts of additives can significantly transfer heat (0.5 mg/mL, DOI: 10.1002/adfm.202308036; 0.1 mg/mL, DOI: 10.1002/sml.202207092).

2) The *o*-CB serves as a template growth agent for perovskite grains, regulating the ordered crystallization of perovskites. The resulting grain boundaries, formed by the interconnection of grains through electron and hole transport layers, serve as channels for heat dissipation, facilitating thermal transfer.

First of all, an exceptional heat transfer ability is claimed in Figure S1. However, this claim lacks context, what is exceptional?

Response: Thank you for reviewing our manuscript carefully. We apologize for the omission of Fig. 1a in the original manuscript. Fig. 1a should be included at the end of this sentence “due to its exceptional heat

transfer ability and chemical stability in the series (Supplementary Fig. 1)”. This ‘exceptional’ illustrates the comparison between *o*-carborane and the other two types of carboranes. *o*-carborane exhibits a lower thermal hysteresis coefficient compared to meta-carborane and para-carborane. A smaller thermal hysteresis indicates better heat dissipation. Additionally, we can observe in Figure S1 that *o*-carborane has a faster cooling rate compared to the hole transport material (PEDOT:PSS), indicating its superior heat transfer capability.

To me the difference between PEDOT:PSS and *o*-CB in Figure S1 appears to be rather small. Please explain and provide a rationale for how this can significantly alter the temperature of the cell (e.g. as shown in Figure 2h).

Response: We thank the further discussion on understanding the role of the cooling rate of *o*-CB to the temperature of the cell. It can be observed from Figure S1 that the cooling rate of *o*-carborane is nearly twice as fast as that of PEDOT:PSS. By comparing the cooling rates of *o*-carborane and PEDOT:PSS, the necessity of introducing carborane is demonstrated. When light enters the device from the side of PEDOT:PSS, energy accumulates at the interface between PEDOT:PSS and perovskite, leading to an increase in temperature at the buried interface. Introducing *o*-carborane at this position can effectively dissipate heat due to its faster

cooling rate and higher thermal conductivity. This helps reduce heat accumulation at the buried interface, accelerate internal temperature cooling, and ultimately lower the operating temperature of the device.

Does Figure 2h hold true for longer illumination, i.e. for operational conditions?

Response: Figure 2h is applicable for longer durations of illumination. From Figure 2e, it can be inferred that the temperature of each layer in the device stabilizes after 1000 s of illumination.

Moreover, while the heat transfer ability has likely an impact on the stability, it is not really proven that this is the main factor for improved device stability. Can the authors prove, that a few degree differences in the temperature of the cell, let's say 4-5 °C (as shown in Figure 2h) can make such a big difference in the stability?

Response: The enhanced thermal stability stems from the following reasons: (1) the improved thermal conductivity of *o*-CB, which suppresses thermal accumulation; (2) the enhanced contact between perovskite and HTL and (3) a more appropriate band alignment at the buried interface, thereby reducing non-radiative recombination and the generation of

phonons (heat). These points have been detailed in the manuscript, collectively contributing to the lowered operating temperature of the solar cells. In addition, the introduction of *o*-carborane also improves the crystallization of perovskites, which also contributes to enhanced thermal stability. All these factors together contribute to the enhanced thermal stability.

Other smaller points to consider for the authors.

How thick were the Films in Figure S1? Related to this how thick is the *o*-CB layer in the device (I guess this can be inferred from Fig. S2)?

Response: We prepared a layer of *o*-CB (1 mg/mL) thin film on glass and conducted atomic force microscope measurements, estimating its thickness to be 5.5 nm (Fig. R6a). The thickness of PEDOT:PSS was estimated to be 56 nm (Fig. R6b). As shown in Supplementary Figure S2, *o*-CB permeates throughout the whole perovskite film but is primarily distributed at the interface between the perovskite and the PEDOT:PSS. Based on this observation, we speculate that the thickness of *o*-CB in the device is approximately several tens of nanometers.

Fig. R6. AFM image and height analysis of a) *o*-CB and b) PEDOT:PSS surface on pure glass.

In Figures 1e and f, SEM images are provided of annealed samples at 85 degrees, please provide in addition original SEM images, this would strengthen the argument.

Response: Thank you for reviewing our manuscript carefully. We have included the SEM images (Fig. R7) taken before annealing and added them to the supporting information (Supplementary Fig. 3).

Fig. R7. SEM of pristine film and treated with *o*-CB before heating at 85 °C.

Regarding Figure 2c and d, please provide in the caption or image the device structure (currently only provided in the main text).

Response: Thank you for reviewing our manuscript carefully. We have supplemented the devices' structure in the caption in the revised manuscript.

Regarding the statement “We then used T-type thermocouples to conduct real-time temperature monitor to investigate the photothermal response of each layer in PSCs” how can this work to detect the temperature of nm thick films? Please explain.

Response: Thanks for the reviewer's question. We utilized the following approach to monitor the temperature of each film layer: a temperature sensor was attached to the backside of the ITO layer, adhering it to the backside to prevent any damage to the film caused by the adhesive. The

temperature response on the film side was then transmitted to the ITO backside for real-time temperature monitoring and determination. The monitoring results showed consistency with the surface temperature displayed by the IR Camera imaging, indicating that this method can be utilized to confirm the surface temperature of the film.

Regarding the importance of electron transfer to *o*-CB, the authors may find a recent study interesting that demonstrated *o*-CB as an interfacial layer at the n-type interface which improved the electron transfer as well (<https://doi.org/10.1038/s41467-022-34203-x>).

Response: We have now cited it in the revised manuscript as ref. 32.

Regarding the energy levels shown in Figure S13, the electron affinity of C_{60} is likely not at 4.5 eV, this would cause too much recombination losses. Related to this, the argument that the energy level offset between PEDOT:PSS and the perovskite gets smaller when the energy level of the perovskite with the *o*-CB shift upwards, this argument would also imply that the C_{60} energy offset gets larger (with current energy levels), please reconsider this statement.

Response: Thank you for reviewing our manuscript carefully and pointing

out our mistake. We did not thoroughly investigate the references and provided incorrect information in the original manuscript. We sincerely apologize for this careless. After careful literature investigation, we found that the electron affinity of C₆₀ is 3.9 eV, and its bandgap is 2 eV. We remade the energy level scheme as shown below in Fig. R8. We also replaced the old one with Fig. R8 (supplementary Fig. 15 in supporting information).

Fig. R8. Energy level arrangement of the complete device.

Reviewer #3 (Remarks to the Author):

This manuscript presents a strategy of introducing ortho-carborane (*o*-CB) to improve both efficiency and stability of mixed Sn-Pb perovskite solar cells and all-perovskite tandem solar cells. The incorporation of *o*-CB effectively transfers heat and lowers the surface temperature of perovskite film by around 5 °C under 1 sun illumination. As a result, the mixed Sn-Pb perovskite solar cell with *o*-CB treatment exhibits improved thermal stability, with 80% of its initial efficiency kept after aging at 85 °C for 1080 hours. Additionally, the introduction of *o*-CB improves the film quality and facilitates the charge transfer, enabling the remarkable efficiencies of 23.44% and 27.2% for mixed Sn-Pb cells and all-perovskite tandem cells, respectively. This work is of great interest and significance to the community of perovskite materials and solar cells, especially in mixed Sn-Pb cells and all-perovskite tandem cells, representing a significant advance. Therefore, I would recommend publication of this work after addressing the following minor concerns issues/comments.

1. Could the authors explain why they considered to use the compound of carboranes?

Response: Thanks for the reviewer's comments. Aiming to improve the

thermal stability of devices through thermal management, we have been seeking materials with high thermal conductivity and specific heat capacity. Carboranes are a class of spherical molecules with delocalized π -orbitals, allowing electrons to move freely, and spread out over a larger region within the materials. The characteristic of electron delocalization enables them to improve thermal conductivity and achieve efficient carrier transport, which can both transfer heat and improve charge transport. It was expected to achieve stable and efficient all-perovskite tandem solar cells.

Compared to other compounds, ortho-carbon borane has a higher specific heat capacity, which is beneficial for improving stability. In terms of device performance, have the other two compounds been tried, and what were the possible results?

Response: We have conducted I - V measurements of devices with different carboranes, and the results are shown in Fig. R9. We can see that devices using meta-carborane and para-carborane exhibit slightly inferior performance compared to devices using ortho-carborane.

Fig. R9. J - V characteristics of the champion devices with different carboranes.

2. “In comparison to the control film which exhibited a localized area of heat accumulation (hotspot), the *o*-CB treated film showed improved heat dissipation and a more uniform temperature distribution.” Can the authors explain why the control film exhibits thermal accumulation in the middle?

Response: Thank you for your comments. The temperature ($T(x)$) at various locations on the perovskite film can be calculated using Eq. 1 (DOI: 10.1016/j.mssp.2015.09.029).

$$T(x) = T_{sub} + \frac{\theta}{\lambda^2} \left(1 - \frac{\sinh \lambda x + \sinh \lambda (1-x)}{\sinh \lambda l} \right) \quad (1)$$

The uniform substrate temperature (T_{sub}) is assumed in case the temperature is constant for all positions. In this equation, θ is constant with position but depends on the temperature. λ is related to the thermal conductivity and the thickness of the perovskite layer. L is the distance

from the edge to the center of the film. Through calculations, the temperature is higher at the center of the film and reduces to the lower values at the edges. From the equation above, we can see that the temperature distribution of the perovskite film depends on several factors, such as the temperature of the substrate, the thermal conductivity of the perovskite film, the thickness of the perovskite film, and so on.

3. “A heat dissipation model (Fig. 2a) with an initial temperature of 25°C and a size of 0.17 cm × 0.15 cm × 0.016 cm was established (see detailed model information 88 in the supplementary materials).” The authors' choice of 1.5 s instead of 0 s as a reference point should be clarified. It would be helpful if the authors could explain the rationale behind this selection. Additionally, it would be interesting to know if the inclusion of 1.5 s has any impact on the overall changes observed in the study.

Response: Thanks for the reviewer's suggestion. Due to the inability to accurately simulate the initial state at 0s during the simulation, and through the T-type thermocouple test, it was found that there was no significant difference in temperature measurements between the control film and the film treated with *o*-CB within the first 1.5s. Therefore, the simulation within 1.5s can accurately represent the same initial temperature point for both the control film and the film treated with *o*-CB. Although *o*-CB, which

has better thermal properties, is added, the overall thermal conductivity of the perovskite film is still lower than that of traditional thermally conductive metals. Therefore, the film still has a low sensitivity to temperature and will not show significant temperature differences during short-term (1.5s) irradiation. Therefore, choosing 1.5s as the initial temperature state can to some extent ensure the accuracy of the results and better reflect that the temperature of the control film increases faster than that of the film treated with *o*-CB after continuous light irradiation under the same initial temperature conditions. This further proves that the film treated with *o*-CB has better thermal conductivity.

4. The introduction of *o*-CB leads to significant reduction of the J - V curve hysteresis of devices. The authors should discuss more to further elaborate on this.

Response: Thanks for the reviewer's suggestion. During the measurement process under illumination conditions, the device experiences a photothermal effect, which leads to a temperature increase (10.1002/adma.201902413). At higher temperatures, charge accumulation occurs, causing higher charge recombination losses, thereby causing the occurrence of hysteresis. The device treated with *o*-CB can reduce charge accumulation under illumination conditions, decreasing charge

recombination losses and thus suppressing the $J-V$ hysteresis to some extent (10.1002/solr.202100370).

5. The authors are suggested to provide more explanations on the statement of “The improved film quality also contributed to reduced Sn oxidation”.

Response: Thanks for the reviewer's suggestion. Several papers have demonstrated this viewpoint. Wang et al. reduced the interface roughness while improving the quality of the tin-based perovskite thin films by modifying the hole transport layer and the perovskite interface in inverted tin-based perovskite cells. This allowed for dimensional distribution regulation and suppression of Sn^{2+} oxidation (DOI: 10.1002/smtd.202300029). Nonuniform perovskite thin film increases the contact of perovskite with water and oxygen in the air, leading to the formation of non-radiative recombination centers and an increased possibility of tin vacancies due to the oxidation of Sn^{2+} to Sn^{4+} (DOI: 10.1021/acsami.0c11253). Therefore, improving the film quality can reduce the erosion of the device by water and oxygen in the air, and inhibit Sn^{2+} oxidation.

6. The authors provide the operational stability of the tandem cells. It is recommended that the authors also analyze and discuss why the thermal

stability of the tandem cells gets improved.

Response: Thanks for the reviewer's suggestion. Owing to the thermal management of *o*-CB, the NBG Sn-Pb perovskite solar cells exhibited better thermal stability as shown in Fig. 4e in the manuscript. Besides, there are I-Br segregation issues in WBG perovskite, which limits the stability of the corresponding perovskite solar cells. After introducing the *o*-CB into WBG perovskite films, the I-Br segregation phenomenon was inhibited (Fig. R10), thereby it can alleviate the damage to the devices under thermal conditions. Therefore, the introduction of *o*-CB is also expected to thermal stability of the tandem cells.

Fig. R10. XRD patterns of the perovskite films a) without and b) with *o*-CB treatment before and after 100 mW cm^{-2} white light illumination for 2 days.

REVIEWER COMMENTS

Reviewer #1 (Remarks to the Author):

The authors have addressed my comments in detail and added some new data including the device performance and stability, but I still have some crucial concerns about the mechanism and effects of the heat transfer model. The major concerns are listed as follows:

1. The response of comment #1 still does not explain that the o-CB device can achieve better heat dissipation in the case of inserting a layer of material and introducing additional thermal resistance. This seems to be contradictory in thermal physics, so please give a clearer mechanism and supporting references, and the main text should also focus on this part.
2. The thermal conductivity of the device shown in Fig. R1 shows that at higher ambient temperatures, the thermal conductivity difference between the device with or without o-CB is gradually reduced. For me the thermal conductivity of the control and target devices at 85 oC can be considered almost no difference. In this case, does the target device really exhibit a lower thermal equilibrium temperature than the control device? If does, it is better to give critical experimental or calculated data, and further clarify whether the temperature difference caused by thermal conductivity is directly related to stability. And if not, does it mean that the equilibrium temperature of the device is determined by the external ambient temperature if the ambient temperature is high enough?
3. The authors mentioned "This allows electrons to move freely, facilitating heat transfer in the entire device" in response 1. The behavior of the electron should be reflected in the TRPL, but the o-CB films seem to be unable to see anything, while the control films show an abnormal charge behavior, which is unreasonable. In addition, the perovskite film on the glass mentioned in the reference (10.1002/aenm.202003489) does not exhibit a sudden decline in a short period of time, but a slow decay. Unless the film quality is extremely poor, the rapid decline of the TRPL curve must be attributed to the extraction behavior based on the transporting layer or electrode. So, the TRPL results need to be further analyze in the case of the two substrates being the same, provided with statistical data or the differential lifetime by mathematical calculation.

Reviewer #2 (Remarks to the Author):

The paper by Tan et al. has further improved during the revision and several points have been clarified. However, there remain a couple of points that need to be better justified in my opinion before the paper should be considered for publication in Nature Communications.

Regarding Supplementary Figure 1., I am still confused, the heat transfer by conduction should dependent inversely on the thickness of the material, and also the mass of the material plays a role. I think the authors need to consider these points and make a fair comparison between PEDOT:PSS and o-CB (e.g. same thickness or mass?). Similarly, regarding the point that o-CB was chosen due to its exceptional thermal hysteresis, I think it would be more appropriate to compare this property to a few other HTLs used for perovskite solar cells, not only with 2 other carboranes. Regarding "It can be observed from Figure S1 that the cooling rate of o-carborane is nearly twice as fast as that of PEDOT:PSS." please specify the cooling rate.

Regarding the listed 3-4 points listed of how o-CB can improve the thermal stability of the device (improved thermal conductivity of o-CB, enhanced contact between perovskite and HTL, better band alignment, improved perovskite crystallization,) and the statement that "All these factors together contribute to the enhanced thermal stability.", I think the authors should implement and discuss these possible explanations in their paper as well. Furthermore, they should explain why these other factors (especially band alignment, and crystallization) do not invalidate their main conclusions about the importance of improved heat transfer through the o-CB.

Lastly, as a small note, in TRPL even with glass you can have an initial charge redistribution/transfer that leads to a decay in the TRPL but I think this should not affect their conclusions based on this measurement here.

Reviewer #3 (Remarks to the Author):

The authors have addressed the comments and issues raised by the reviewers. I am satisfied with their response and revision. I would recommend acceptance of this manuscript at its current version.

Reviewer #1 (Remarks to the Author):

The authors have addressed my comments in detail and added some new data including the device performance and stability, but I still have some crucial concerns about the mechanism and effects of the heat transfer model. The major concerns are listed as follows:

1. The response of comment #1 still does not explain that the *o*-CB device can achieve better heat dissipation in the case of inserting a layer of material and introducing additional thermal resistance. This seems to be contradictory in thermal physics, so please give a clearer mechanism and supporting references, and the main text should also focus on this part.

Response: We apologize for confusing the reviewer with our unclear explanation in the text. Through SIMS measurements, it is found that *o*-CB is distributed in a gradient within the perovskite film, decreasing gradually from the bottom buried interface upwards. It is distributed as an additive within the perovskite, not as a separate layer sandwiched between the hole transport layer and the perovskite layer. However, since *o*-CB shows high thermal conductivity, the introduction of *o*-CB reduces the overall thermal resistance (R) and increases the thermal transmissivity (K) of the device, as shown in Table R1. We can see that after the introduction of *o*-CB, the device's thermal resistance at 25°C, 55°C, and 85°C is lower than that of the control device at each temperature, while the thermal transmissivity is higher than that of the control device at each temperature, thus achieving better heat dissipation.

Table R1. The thermal resistance (R) and thermal transmissivity (K) of devices w/ and w/o *o*-CB.

Control	R (°C/W)	K (W/mk)	With o-CB	R (°C/W)	K (W/mk)
25°C	2.1782	1.4430	25°C	2.1167	1.4780

55°C	2.2031	1.4200	55°C	2.1379	1.4350
85°C	2.2225	1.3550	85°C	2.1641	1.3620

More discussion was added in the revised manuscript:

“In addition, we conducted tests on the thermal resistance and thermal transmissivity of the entire device, which provide insights into the device's heat conduction capacity. We can see from Supplementary Table 2 that after the introduction of *o*-CB, the device's thermal resistance at 25°C, 55°C, and 85°C is lower than that of the control device at each temperature, while the thermal transmissivity is higher than that of the control device at each temperature, thus achieving better heat dissipation.”

2. The thermal conductivity of the device shown in Fig. R1 shows that at higher ambient temperatures, the thermal conductivity difference between the device with or without *o*-CB is gradually reduced. For me the thermal conductivity of the control and target devices at 85 °C can be considered almost no difference. In this case, does the target device really exhibit a lower thermal equilibrium temperature than the control device? If does, it is better to give critical experimental or calculated data, and further clarify whether the temperature difference caused by thermal conductivity is directly related to stability. And if not, does it mean that the equilibrium temperature of the device is determined by the external ambient temperature if the ambient temperature is high enough?

Response: Thank the reviewer for the careful review of our manuscript. The test results of the thermal conductivity of the devices demonstrated that as the temperature increased, the difference in thermal conductivity between the devices with *o*-CB and the control devices gradually decreased. The thermal equilibrium temperature not only depends on the thermal conductivity but also on the heat generation rate and heat dissipation rate. We speculate that the introduction of *o*-CB will lower the device's thermal equilibrium temperature. To confirm this hypothesis, we placed the control device and the device with *o*-CB on a hot plate at 85 °C and tested the temperature

changes of the device using an infrared camera. The temperature changes of the devices over time are shown in Fig. R1. We can see that as time progresses, the temperature of both devices increases, but the temperature of the control device increases at a faster rate. After 2000 seconds, the thermal equilibrium temperature of the control device is noticeably higher than that of the device with *o*-CB. Higher thermal conductivity is beneficial for heat transfer and thus enhances the stability of the device.

Fig. R1. Comparison of the heating up process of devices w/ and w/o *o*-CB through IR thermal images. These pictures were taken from the glass side.

3. The authors mentioned “This allows electrons to move freely, facilitating heat transfer in the entire device” in response 1. The behavior of the electron should be reflected in the TRPL, but the *o*-CB films seem to be unable to see anything, while the control films show an abnormal charge behavior, which is unreasonable. In addition, the perovskite film on the glass mentioned in the reference (10.1002/aenm.202003489) does not exhibit a sudden decline in a short period of time, but a slow decay. Unless the film quality is extremely poor, the rapid decline of the TRPL curve must be attributed to the extraction behavior based on the transporting layer or electrode. So, the TRPL results need to be further analyze in the case of the two substrates being the same, provided with statistical data or the differential lifetime by mathematical calculation.

Response: According to the reviewer's suggestion, we calculated the differential lifetime of both perovskite films w/ and w/o *o*-CB on the glass to describe the PL decay.

The differential lifetime (τ_{PL}) can be estimated in terms of the following Equation to interpret the PL decay,

$$\tau_{PL} = \left(-\frac{1}{m} \frac{d \ln(\Phi_{PL})}{dt} \right)^{-1}$$

where $\Phi_{PL}(t)$ is the PL intensity at t after the photoexcitation, and m is a factor in relation to the injection level, which is set as 2 in this work. By solving the Equation, the results are shown in Fig. R2 by plotting τ_{PL} as a function of the logarithm of PL intensity $\ln(\Phi_{PL})$. According to the platform part of the data result curve, we extracted the effective Shockley reading Hall (SRH) lifetime in the bulk (glass/perovskite) of $\tau_{SRH}^{o-CB} = 3.19 \mu\text{s}$ and $\tau_{SRH}^{Ctrl} = 278.6 \text{ ns}$. Depending on the defect concentration, SRH lifetime is related to the defect-mediated recombination in perovskite bulk. It demonstrates that perovskite film containing *o*-CB has a higher SRH lifetime. The reduction of the SRH recombination center of the film with *o*-CB treatment ushered in a higher SRH lifetime, which also proved that the defect density of the film after *o*-CB treatment was lower.

Fig. R2. Differential lifetime versus the logarithm of the PL intensity ($\ln(\Phi_{PL})$).

More discussion was added in the revised manuscript:

“We further described the PL decay using the differential lifetime (τ_{PL}), which can be estimated in terms of the following Equation, $\tau_{PL} = \left(-\frac{1}{m} \frac{d \ln(\Phi_{PL})}{dt} \right)^{-1}$, where $\Phi_{PL}(t)$ is the PL intensity at t after the photoexcitation, and m is a factor in relation to the injection

level, which is set as 2 in this work. By solving the Equation, the results are shown in Supplementary Fig.18 by plotting τ_{PL} as a function of the logarithm of PL intensity $\ln(\Phi_{PL})$. According to the platform part of the data result curve, we extracted the effective Shockley reading Hall (SRH) lifetime in the bulk (glass/perovskite) of $\tau_{SRH}^{o-CB}=3.19 \mu s$ and $\tau_{SRH}^{Ctrl}=278.6 ns$. Depending on the defect concentration, SRH lifetime is related to the defect-mediated recombination in perovskite bulk. It demonstrates that perovskite film containing *o*-CB has a higher SRH lifetime. The reduction of the SRH recombination center of the film with *o*-CB treatment ushered in a higher SRH lifetime, which also proved that the defect density of the film after *o*-CB treatment was lower.”

Reviewer #2 (Remarks to the Author):

The paper by Tan et al. has further improved during the revision and several points have been clarified. However, there remain a couple of points that need to be better justified in my opinion before the paper should be considered for publication in Nature Communications.

Regarding Supplementary Figure 1., I am still confused, the heat transfer by conduction should dependent inversely on the thickness of the material, and also the mass of the material plays a role. I think the authors need to consider these points and make a fair comparison between PEDOT:PSS and *o*-CB (e.g. same thickness or mass?). Similarly, regarding the point that *o*-CB was chosen due to its exceptional thermal hysteresis, I think it would be more appropriate to compare this property to a few other HTLs used for perovskite solar cells, not only with 2 other carboranes. Regarding "It can be observed from Figure S1 that the cooling rate of *o*-carborane is nearly twice as fast as that of PEDOT:PSS." please specify the cooling rate.

Response: Thanks for the reviewer's suggestion. According to the reviewer's suggestion,

we made *o*-CB film and PEDOT:PSS film based on the same concentration, along with the other two most commonly used hole transport materials (PTAA and Me-4PAZC) in p-i-n devices. we tested and calculated the cooling rate of *o*-CB, PEDOT:PSS, PTAA, and Me-4PAZC (with the same concentration of 1.3 wt%). The experimental results are shown in Fig. R3. It can be seen that the cooling rate of *o*-CB is 0.5 °C/s, which is higher than that of the other three materials. We updated Supplementary Figure 1 in the revised manuscript.

Fig. R3. Variation of surface temperature with time for *o*-CB (1.3 wt%) 、 Me-4PACZ (1.3 wt%) 、 PTAA (1.3 wt%) and PEDOT:PSS (1.3 wt%) from 100 °C to 25 °C. Cooling rate: PEDOT:PSS_{rate} = 0.341 °C/s; PTAA_{rate} = 0.3125 °C/s ; Me-4PACZ_{rate} = 0.31 °C/s; *o*-CB_{rate} = 0.5 °C/s.

Regarding the listed 3-4 points listed of how *o*-CB can improve the thermal stability of the device (improved thermal conductivity of *o*-CB, enhanced contact between perovskite and HTL, better band alignment, improved perovskite crystallization,) and the statement that "All these factors together contribute to the enhanced thermal stability.", I think the authors should implement and discuss these possible explanations in their paper as well. Furthermore, they should explain why these other factors (especially band alignment, and crystallization) do not invalidate their main conclusions about the importance of improved heat transfer through the *o*-CB.

Response: Thanks for the reviewer's suggestion. According to the reviewer's suggestion, we conducted further analysis on factors that may improve the thermal stability of the device. To demonstrate that the improved crystallization of perovskite does not invalidate the increase of thermal conductivity introduced by *o*-CB, leading to improved thermal stability of the device, we directly dropped the precursor solution with/without *o*-CB onto the substrate. We then placed them on a 100 °C hot plate for 15 minutes to allow the excess solvent to completely evaporate. After annealing, we connected the thin film with a T-type thermocouple and placed them under 100 mW cm⁻² of AM 1.5G light for temperature tracking. The results are shown in Fig. R4. For poorly crystalline films, the pristine perovskite film exhibited a continuously higher temperature than the *o*-CB-treated film upon heating.

More discussion was added in the revised manuscript:

“The enhanced thermal stability might stem from the following reasons: 1) the improved thermal conductivity of *o*-CB, which suppresses thermal accumulation;³³ 2) the enhanced contact between perovskite and HTL and 3) a more appropriate band alignment at the buried interface,³⁴ thereby reducing non-radiative recombination and the generation of phonons (heat). In addition, the introduction of *o*-CB also improves the crystallization of perovskite, which also contributes to enhanced thermal stability.³⁵ To demonstrate that the improved crystallization of perovskite does not invalidate the increase of thermal conductivity introduced by *o*-CB, leading to improved thermal stability of the device, we directly dropped the precursor solution with/without *o*-CB onto the substrate. We then placed them on a 100 °C hot plate for 15 minutes to allow the excess solvent to completely evaporate. After annealing, we connected the thin film with a T-type thermocouple and placed them under 100 mW cm⁻² of AM 1.5G light for temperature tracking. The results are shown in Supplementary Fig. 22. For poorly crystalline films, the pristine perovskite film exhibited a continuously higher temperature than the film treated by *o*-CB upon heating.”

Fig. R4. Continuous recording of the temperature change of perovskite films w/ and w/o *o*-CB treatment under illumination (AM 1.5G, 100 mW cm⁻²).

Lastly, as a small note, in TRPL even with glass you can have an initial charge redistribution/transfer that leads to a decay in the TRPL but I think this should not affect their conclusions based on this measurement here.

Response: Thanks for the reminder, we agree with what the reviewer said. To further describe the PL decay, we calculated the differential lifetime of both perovskite films w/ and w/o *o*-CB on the glass. The Shockley reading Hall (SRH) lifetime in the bulk (glass/perovskite) of $\tau_{SRH}^{o-CB}=3.19 \mu s$ and $\tau_{SRH}^{Ctrl}=278.6 ns$ were obtained, which is related to the defect-mediated recombination in perovskite bulk. The perovskite film containing *o*-CB has a higher SRH lifetime, indicating that the defect density of the film after *o*-CB treatment was lower than the pristine film.

Reviewer #3 (Remarks to the Author):

The authors have addressed the comments and issues raised by the reviewers. I am satisfied with their response and revision. I would recommend acceptance of this manuscript at its current version.

Response: Thank you for recommending the acceptance of our manuscript.

REVIEWERS' COMMENTS

Reviewer #1 (Remarks to the Author):

I am satisfied with the revisions.

Reviewer #2 (Remarks to the Author):

Ok, the manuscript has further improved. For Supplementary Figure 1, please add that used concentration in the caption. Regarding my other point about the alternative explanations, I would suggest writing this a bit more clearly "To demonstrate that the improved crystallization of perovskite does not invalidate the increase of thermal conductivity introduced by o-CB, leading to improved thermal stability of the device". I was mostly concerned that there are in principle many factors (e.g. enhanced contact between perovskite and HTL, better band alignment, improved perovskite crystallization) that can lead to improved stability, and it's hard to assign it conclusively to the enhanced thermal conductivity (which is the main point here). However, it is partially acknowledged. "The enhanced thermal stability might stem from the following reasons" so I am okay with it. The authors can check if any claims need to be slightly adjusted along these lines.

Reviewer #1 (Remarks to the Author):

I am satisfied with the revisions.

Response: Thank the reviewer for approving our revision.

Reviewer #2 (Remarks to the Author):

Ok, the manuscript has further improved. For Supplementary Figure 1, please add that used concentration in the caption.

Response: According to the suggestion of the reviewer, we have added the concentrations in the caption of Supplementary Figure 1 in the revised supporting information.

Regarding my other point about the alternative explanations, I would suggest writing this a bit more clearly "To demonstrate that the improved crystallization of perovskite does not invalidate the increase of thermal conductivity introduced by o-CB, leading to improved thermal stability of the device". I was mostly concerned that there are in principle many factors (e.g. enhanced contact between perovskite and HTL, better band alignment, improved perovskite crystallization) that can lead to improved stability, and it's hard to assign it conclusively to the enhanced thermal conductivity (which is the main point here). However, it is partially acknowledged. "The enhanced thermal stability might stem from the following reasons" so I am okay with it. The authors can check if any claims need to be slightly adjusted along these lines.

Response: Thank the reviewer for the suggestions, we have made adjustments in the revised manuscript. The revised section has been highlighted in yellow.